# HyperSHAP: Shapley Values and Interactions for Hyperparameter Importance

## Abstract

Hyperparameter optimization (HPO) is a crucial step in achieving strong predictive performance, particularly for deep learning with hyperparameters controlling the neural architecture and learning behavior. However, the impact of some hyperparameters on model generalization can vary significantly depending on the dataset and performance measure, making it challenging to generalize their importance. Gaining a better understanding of the importance of hyperparameters is therefore important to deepen our understanding of machine learning and to leverage this knowledge in future downstream HPO tasks, especially if training is expensive and HPO needs to be as efficient as possible. To address these challenges, we propose a game theoretic framework based on Shapley values and interactions for HPO. These methods offer an additive decomposition of a performance measure across hyperparameters, enabling both local and global explanations of hyperparameter importance and interactions. Our framework, named HyperSHAP, provides insights into ablation studies, tunability of specific hyperparameter configurations, and entire configuration spaces. Through experiments, we demonstrate that focusing on the hyperparameters deemed important by our framework can improve performance during subsequent hyperparameter optimization, while ignoring important hyperparameters or interactions degrades performance. This validates the effectiveness of our approach in enhancing model performance and providing meaningful, interpretable explanations of hyperparameter importance.

## 1 Introduction

Hyperparameter optimization is an indispensable step in the design process of developing machine learning applications to achieve the best possible performance for a given dataset and performance measure (Bischl et al., 2023). This is particularly true for deep learning methods exposing different kinds of hyperparameters for describing the neural architecture and steering the learning behavior, e.g., the learning rate or weight decay (Zimmer et al., 2021). Moreover, through the advent of generative AI and large language models that are fine-tuned to various tasks, hyperparameter optimization is key to achieving the best-possible results (Yin et al., 2021; Tribes et al., 2023; Wang et al., 2023).

However, different hyperparameters affect the generalization performance of the resulting model differently (Bergstra & Bengio, 2012; Hutter et al., 2014; Zimmer et al., 2021) and, thereby, their importance to be tuned. Moreover, the importance of hyperparameters that lead to an effective improvement in generalization performance can depend on the dataset and performance measure of interest (Bergstra & Bengio, 2012; van Rijn & Hutter, 2018). Due to this, so far, it is difficult to draw general conclusions about the importance of individual hyperparameters to better understand the corresponding learning algorithm. In addition, just observing an optimized hyperparameter configuration, it is difficult to attribute the effect of individual hyperparameter values to the improvement in generalization performance.

To address this issue, methods for eliciting hyperparameter importance have been devised, ranging from local methods to determine the effect of single hyperparameter values over a default value (Fawcett & Hoos, 2016; Biedenkapp et al., 2017) to global explanations performing a symbolic regression (Segel et al., 2023) or a variance decomposition (Hutter et al., 2014; Watanabe et al., 2023; Theodorakopoulos et al., 2024). While the former can only detect main effects and ignores

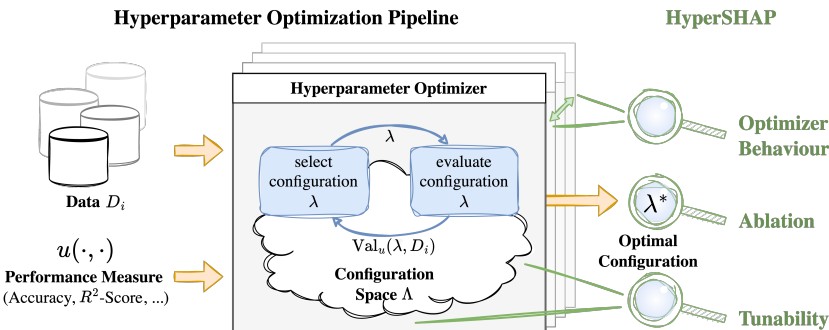

Figure 1: Game-theoretic explanations as defined with HYPERSHAP's hyperparameter importance games can be used to gain insights into hyperparameter values, hyperparameter configuration spaces, datasets, and different hyperparameter optimizers (frames). HYPERSHAP can be used for data-specific explanations or explanations across datasets.

interactions between hyperparameter values, the latter can take interactions into account to a limited degree but is rather complicated to implement, and its result is rather difficult to interpret.

In this paper, we propose HYPERSHAP, a post-hoc explanation framework for hyperparameter importance and interactions between hyperparameters based on Shapley values (Shapley, 1953) and Shapley interaction indices (Tsai et al., 2023). Stemming from the field of algorithmic game theory, Shapley values represent an additive decomposition of a given value function, in our case, a performance measure, across a set of players, in our case (values of) hyperparameters. Overall, we define 5 games to quantify the importance of hyperparameters, each of which can be used to obtain different types of explanations either of a given hyperparameter configuration, of a hyperparameter search space, or an optimizer's characteristics.

While we generate first insights with the help of our framework, we also showcase the usefulness of the explanations in a downstream task, performing hyperparameter optimization for the same dataset and performance measure as a proof of concept that our approach identifies, in fact, meaningful importances and interactions. In our experiments, focusing on hyperparameters identified as important with a low degree of interactions by HYPERSHAP proves beneficial, resulting in better anytime performance. However, ignoring the presence of interactions deteriorates performance in turn.

**Contribution.** All in all, our contributions are threefold:

(1) First, we propose HYPERSHAP, a novel framework for determining hyperparameter importance based on Shapley additive explanations and interaction indices. Therewith, we define 5 different games that can be considered for obtaining explanations on three levels: specific configurations, hyperparameter spaces, and optimizer behavior.

(2) Second, with the help of HYPERSHAP, we elicit hyperparameter importance and interaction structures for `lcnet` (Zimmer et al., 2021), observing that while higher-order interactions among the hyperparameters exist, considering only lower-order interactions is typically sufficient to explain most of the performance improvements.

(3) Third, we demonstrate in a downstream task the practical usefulness of these explanations. In particular, we showcase how focusing on important hyperparameters, and thereby reducing the search space, improves anytime performance of hyperparameter optimizers in constrained budget settings.

## 2 HYPERPARAMETER OPTIMIZATION AND IMPORTANCE

Hyperparameter optimization (HPO) is concerned with the problem of finding the most suitable hyperparameter configuration of a learner $A$ for a given task, typically consisting of some (training) dataset $D$ and some performance measure $u$ quantifying the usefulness (Bischl et al., 2023). To

put it more formally, let $\mathcal{X}$ be an instance space and $\mathcal{Y}$ a label space and suppose $x \in \mathcal{X}$ are (non-deterministically) associated with labels $y \in \mathcal{Y}$ via a joint probability distribution $P(\cdot, \cdot)$. Then, a training dataset $D = \{(x^{(k)}, y^{(k)})\}_{k=1}^{N} \subset \mathcal{X} \times \mathcal{Y}$ is a sample from that probability distribution. Furthermore, a predictive performance measure $u : \mathcal{Y} \times P(\mathcal{Y}) \to \mathbb{R}$ is a function mapping tuples consisting of a label and a probability distribution over the label space to the reals. A learner $A$ can be understood as a function $A_{\boldsymbol{\Lambda}} : \mathbb{D} \to \mathcal{H}$ that is parameterized by some hyperparameter space $\boldsymbol{\Lambda}$ and maps datasets from the dataset space $\mathbb{D}$ to a corresponding hypothesis space $\mathcal{H} := \{h \mid h : \mathcal{X} \to P(\mathcal{Y})\}$.

As the parameterization $\boldsymbol{\lambda} \in \boldsymbol{\Lambda}$ of $A$ typically impacts the hypothesis space $\mathcal{H}$ and biases the learning behavior in some way, it needs to be tuned to the task at hand, i.e., to the given dataset and loss function. The task of HPO is then to find a parameterization that yields a hypothesis that generalizes well beyond the data used for training. For brevity, we denote the hypothesis obtained through applying $A_{\boldsymbol{\lambda}}$ to dataset $D$ by $h_{\boldsymbol{\lambda}, D}$. For a given dataset $D \in \mathbb{D}$, then the following optimization problem needs to be solved:

$$\boldsymbol{\lambda}^{*} \in \arg\max_{\boldsymbol{\lambda} \in \boldsymbol{\Lambda}} \int_{(x,y) \sim P(\cdot, \cdot)} u\big(y, h_{\boldsymbol{\lambda}, D}(x)\big) \ .$$

As the true generalization performance is intractable, it is estimated by splitting the given dataset $D$ into training $D_{train}$ and validation data $D_{val}$. Accordingly, we obtain

$$\boldsymbol{\lambda}^{*} \in \arg\max_{\boldsymbol{\lambda} \in \boldsymbol{\Lambda}} \text{VAL}_u(\boldsymbol{\lambda}, D),$$

$$\text{with } \text{VAL}_u(\boldsymbol{\lambda}, D) := \mathbb{E}_{(D_{train}, D_{val}) \sim D} \left[ \frac{1}{|D_{val}|} \sum_{(x,y) \in D_{val}} u\big(y, h_{\boldsymbol{\lambda}, D_{train}}(x)\big) \right] \ .$$

Naively, hyperparameter optimization can be approached by discretizing the domains of hyperparameters and conducting a grid search or by a random search (Bergstra & Bengio, 2012). However, as both these approaches are neither very effective nor efficient, more sophisticated tools have been developed in two directions to increase the efficiency of sampling and evaluation, mainly based on Bayesian and multi-fidelity optimization (Bischl et al., 2023).

**Hyperparameter Importance.** From an intuitive perspective, it is quite obvious that different types of hyperparameters can be of different importance and that their effect also depends on the dataset at hand. However, determining the effect of every hyperparameter requires additional tools, e.g., eliciting the individual importance of a hyperparameter via ablations (Biedenkapp et al., 2017). Additionally, hyperparameters may also influence the effect of other hyperparameters, which is why the importance of a hyperparameter $\lambda_j$ is typically specified as its performance-induced variance (Jin, 2022).Accordingly, computing marginals for quantifying the effect of single hyperparameters as well as their interactions with other hyperparameters can be used to determine the importance of hyperparameters in the functional ANOVA framework (Hutter et al., 2014).

## 3 RELATED WORK

Hyperparameter importance has garnered significant attention in machine learning as it plays a crucial role in providing evidence for the need for tuning hyperparameters and in attributing performance improvements through hyperparameter optimization to hyperparameters (Probst et al., 2019; Pushak & Hoos, 2020; 2022; Schneider et al., 2022). A variety of approaches have been developed to assess how different hyperparameters affect the performance of resulting models, ranging from simple (surrogate-based) ablations (Biedenkapp et al., 2017) to sensitivity analyses and eliciting interactions between hyperparameters based on the functional ANOVA framework (Hutter et al., 2014; van Rijn & Hutter, 2018; Watanabe et al., 2023; Bahmani et al., 2021). Similarly, Moosbauer et al. (2021) propose partial dependence plots visualizing variance across the domain of hyperparameters. In this work, we propose an alternative way of quantifying the importance of hyperparameters with the help of Shapley values and focus on eliciting interactions between hyperparameters with the help of SHAP interaction indices. We focus on the quantification of interactions since, in previous works, it has been noticed that interaction is occasionally comparably low (Zimmer et al., 2021; Novello

et al., 2023), which could serve as a foundation for a new generation of HPO methods that do not assume interactions to be omnipresent.

Beyond quantifying hyperparameter importance, to better understand the impact of hyperparameters and the tuning behavior of hyperparameter optimizers, other approaches have been proposed, such as algorithm footprints (Smith-Miles & Tan, 2012) or deriving symbolic explanations (Segel et al., 2023), functioning as an interpretable model for estimating the performance of a learner from its hyperparameters. In this work, we focus on quantifying the impact of tuning a hyperparameter on the performance.

## 4 SHAPLEY VALUES AND INTERACTION INDICES

Cooperative game theory has been widely applied in machine learning to assign contributions of entities, such as features or data points for a given task (Rozemberczki et al., 2022). Most prominently, to interpret predictions of black box models using feature attributions (Lundberg & Lee, 2017), or to quantify the value of data points (Ghorbani & Zou, 2019). Shapley Interactions (SIs) (Grabisch & Roubens, 1999) extend the Shapley Value (SV) by additionally assigning contributions to groups of entities, which reveal *synergies and redundancies*. Feature interactions uncover additive structures in predictions, which are necessary to understand complex decisions (Lundberg et al., 2020; Sundararajan et al., 2020; Tsai et al., 2023; Bordt & von Luxburg, 2023). The SV and SIs are defined based on a cooperative game comprising $n$ players $\mathcal{N} = \{1, \ldots, n\}$ and $\nu : 2^{\mathcal{N}} \to \mathbb{R}$ as a real-valued set function on the powerset $2^{\mathcal{N}}$. This game captures a joint payout $\nu(S)$ obtained from a set of players $S \subseteq N$ forming a coalition. In this section, we assume $\nu(\emptyset) = 0$ for readability, which does not affect the SV and SIs due to the dummy axiom (introduced below). Given $\nu$, the SV (Shapley, 1953) is the *fair* contribution of an individual entity to the overall payout $\nu(\mathcal{N})$. The SV is uniquely characterized by four intuitive axioms: *linearity* (contributions are linear for linear combination of games), *symmetry* (players with equal contributions obtain equal payout), *dummy* (players that do not change the payout receive zero payout), and *efficiency* (the sum of all payouts equals the joint payout). The SV $\phi^{\mathrm{SV}}(i)$ of player $i \in \mathcal{N}$ can be computed as a weighted average

$$\phi^{\mathrm{SV}}(i) := \sum_{T \subseteq \mathcal{N} \setminus \{i\}} \frac{1}{n \cdot \binom{n-1}{|T|}} \Delta_i(T) \text{ over marginal contributions } \Delta_i(T) := \nu(T \cup i) - \nu(T).$$

Due to the efficiency axiom, the sum of SVs yields the joint payout $\nu(\mathcal{N}) = \sum_{i \in \mathcal{N}} \phi^{\mathrm{SV}}(i)$. Moreover, any game value can be approximated with $\hat{\nu}(T) = \sum_{i \in T} \phi^{\mathrm{SV}}(i)$, which is the best approximation of $\nu$ restricted to individual contributions in terms of a particular optimization objective (Charnes et al., 1988). Clearly, individual contributions are limited in describing the values $\nu(S)$ for every subset $S \subseteq \mathcal{N}$. The Möbius Interactions (MIs) $m : 2^{\mathcal{N}} \to \mathbb{R}$, alternatively Möbius transform (Rota, 1964) or Harsanyi dividend (Harsanyi, 1963), recover any game value additively as

$$\nu(T) = \sum_{S \subseteq T} m(S) \text{ with } m(S) := \sum_{T \subseteq S} (-1)^{|S|-|T|} \nu(T) \text{ for all } S, T \subseteq \mathcal{N}. \tag{1}$$

In fact, the MIs are the unique measure with this property (Harsanyi, 1963). The MI $m(S)$ captures the *pure additive contribution* that is achieved by forming the coalition $S$, which cannot be attributed to any subgroup of players in $S$. Moreover, the MIs form a basis of the vector space of games (Grabisch, 2016), and the SV can be derived as

$$\phi^{\mathrm{SV}}(i) = \sum_{S \subseteq \mathcal{N} : i \in S} \frac{1}{|S|} m(S) \text{ for all } i \in \mathcal{N}.$$

In other words, the SV summarizes all MIs $m(S)$ by equally distributing the interaction among the players in $S$. While the SV has limited expressivity, the MIs with $2^n$ components are difficult to interpret. As a remedy, SIs, $\Phi_k$, provide a flexible framework with interactions for subsets up to size $k = 1, \ldots, n$, where the edge cases are the SV ($k = 1$) and the MIs ($k = n$). Higher-order SIs are computed based on discrete derivatives $\Delta_S(T)$, as an extension of marginal contributions. For two players $i, j \in \mathcal{N}$, the discrete derivative $\Delta_{\{i,j\}}(T)$ of $\{i, j\}$ in the presence of $T \subseteq \mathcal{N} \setminus \{i, j\}$ is given as $\nu(T \cup \{i, j\}) - \nu(T) - \Delta_{\{i\}}(T) - \Delta_{\{j\}}(T)$, i.e., the effect of adding $\{i, j\}$ jointly

minus their individual marginal contributions. This recursion can be extended to any subset $S \subseteq N$, yielding $\Delta_S(T)$. The Shapley Interaction Index (SII) (Grabisch & Roubens, 1999), as an axiomatic extension of the SV to all subsets $S \subseteq \mathcal{N}$ is then a weighted average

$$\phi^{\text{SII}}(S) = \sum_{T \subseteq \mathcal{N} \setminus S} \frac{1}{(n - |S| + 1) \cdot \binom{n-|S|}{|T|}} \Delta_S(T) \text{ over } \Delta_S(T) := \sum_{L \subseteq S} (-1)^{|S|-|L|} \nu(T \cup L).$$

A positive interaction indicates a synergistic effect, whereas a negative interaction indicates redundancies (on average). Given an *explanation order* $k$, the $k$-Shapley Values ($k$-SIIs), $\Phi_k^{\text{SII}}$ (Lundberg et al., 2020; Bordt & von Luxburg, 2023), are recursively constructed from the SII, such that the highest order coincides. Alternatively, the Faithful Shapley Interaction Index (FSII), $\Phi_k^{\text{FSII}}$ (Tsai et al., 2023), constructs SIs based on the best $k$-additive approximation $\hat{\nu}_k(S) := \sum_{L \subseteq S: |L| \leq k} \Phi_k(L)$ of $\nu(S)$ across all subsets $S$ weighted by the Shapley kernel (Tsai et al., 2023), cf. Appendix C.2. FSII are thus well-suited to analyze the degree of interactions within a game. In general, SIs differ in weighting of discrete derivatives (Fumagalli et al., 2023) and MIs of order larger than $k$ (Bordt & von Luxburg, 2023). The SIs provide a flexible framework to adjust explanation expressivity and complexity based on practitioner needs (Tsai et al., 2023; Fumagalli et al., 2024). In the following, we explore interactions within hyperparameter optimization using SIs.

## 5 HYPERSHAP: HYPERPARAMETER IMPORTANCE GAMES

In hyperparameter optimization, a wide variety of questions can be asked for explanations, ranging from individual suggested values in a returned hyperparameter configuration to complex reasoning during the optimization process or the description of remaining optimization potentials. Hence, explanations may be needed on different levels of the hyperparameter optimization process, ranging from returned configurations to a qualitative comparison of entire hyperparameter optimization tools. In the following, we will limit ourselves to three areas, dubbed Ablation, Tunability, and Optimizer Bias, which we describe in more detail in Sections 5.1, 5.2, and 5.3, respectively.

### 5.1 ABLATION BETWEEN TWO HYPERPARAMETER CONFIGURATIONS

One common scenario for quantifying the importance of hyperparameters is to compare a hyperparameter configuration (HPC) $\boldsymbol{\lambda}^*$ of interest to some reference HPC $\boldsymbol{\lambda}^0$, e.g., the default parameterization of a learner as provided by its implementing library or a tuned default HPC that has proven effective for past tasks. In turn, $\boldsymbol{\lambda}^*$ can be an HPC returned by a hyperparameter optimizer or a manually configured HPC. Given $\boldsymbol{\lambda}^*$ and $\boldsymbol{\lambda}^0$, the question now is how values of $\boldsymbol{\lambda}^*$ affect the performance of the learner relative to the reference HPC $\boldsymbol{\lambda}^0$. To this end, we can transition from the reference HPC to the HPC of interest by switching the values of hyperparameters one by one from its value in $\boldsymbol{\lambda}^0$ to the value in $\boldsymbol{\lambda}^*$, which is also done in empirical machine learning studies and referred to as ablations.

While a similar approach has already been followed by Fawcett & Hoos (2016) and Biedenkapp et al. (2017), it fails to quantify interactions between hyperparameters. However, with the help of SVs and SIs, we can overcome this limitation and define the hyperparameter importance game of Ablation as follows.

**Definition 1** (HPI Game - Ablation)**.** *The Ablation HPI game is defined as a tuple*

$$G_A = (\mathcal{N}, \boldsymbol{\lambda}^0, \boldsymbol{\lambda}^*, D, \nu),$$

*consisting of a player set $\mathcal{N}$, a reference HPC $\boldsymbol{\lambda}^0$, an HPC of interest $\boldsymbol{\lambda}^*$, a dataset $D$, and a value function $\nu$. Given a coalition $S \subseteq \mathcal{N}$, we construct an intermediate HPC $\boldsymbol{\lambda}^S$ from $\boldsymbol{\lambda}^0$ and $\boldsymbol{\lambda}^*$ as:*

$$\boldsymbol{\lambda}^S = \begin{cases} \lambda_i^* & \text{if } i \in S \\ \lambda_i^0 & \text{else} \end{cases}$$

*Then, the value function $\nu : 2^{\mathcal{N}} \to \mathbb{R}$ is defined as*

$$\nu(S) = \text{VAL}_u(\boldsymbol{\lambda}^S, D) \ .$$

Table 1: Overview of games to compute hyperparameter importance across three levels: configuration, hyperparameter configuration space, and at the level of the optimizer's behavior.

| | **Ablation** | **Tunability** | **Optimizer Bias** |
|---|---|---|---|
| **Player** | A player represents a hyperparameter value. | A player represents the domain of a hyperparameter. | A player represents the domain of a hyperparameter. |
| **Coalition** | A coalition denotes which hyperparameters' values are switched from its value of the reference HPC $\boldsymbol{\lambda}^0$ to its value in $\boldsymbol{\lambda}^*$. | A coalition determines a subset of hyperparameters tuned by a hyperparameter optimizer. | A coalition determines a subset of hyperparameters tuned by the ensemble of hyperparameter optimizers. |
| **Value Function** | The value function measures the performance of the learner parameterized with the intermediate hyperparameter configuration $\boldsymbol{\lambda}^S$, switch values from the ones given in $\boldsymbol{\lambda}^0$ to $\boldsymbol{\lambda}^*$. | The value function measures the maximum achievable performance of an optimizer considering the configuration space as specified by the coalition. | The value function measures the difference between the best performances found by some optimizer and by any member of an optimizer ensemble. |

## 5.2 Tunability of Learners

Zooming out from a specific configuration, we can ask to what extent it is worthwhile to tune hyperparameters. In the literature, this question has been connected to the term of *tunability* (Probst et al., 2019). Tunability aims to quantify how much performance improvements can be obtained by tuning a learner comparing against a reference parameterization, e.g., a parameterization that is known to work well across various datasets (Pushak & Hoos, 2020).

**Definition 2** (HPI Game - Tunability). *The tunability HPI game is defined as a tuple*

$$G_T = (\mathcal{N}, \boldsymbol{\lambda}^0, \boldsymbol{\Lambda}, \mathcal{D}, \nu),$$

*consisting of a set of players $\mathcal{N}$, a reference HPC $\boldsymbol{\lambda}^0 \in \boldsymbol{\Lambda}$, a hyperparameter configuration space $\boldsymbol{\Lambda}$, a collection of datasets $\mathcal{D} = \{D_1, D_2, \ldots D_M\}$, and a value function $\nu$.*

*Given a coalition $S$, we construct a hyperparameter configuration space $\boldsymbol{\Lambda}^S$ from the original $\boldsymbol{\Lambda}$ as a subspace $\boldsymbol{\Lambda}^S = \times_{i \in S} \Lambda_i$ and the value function is defined as*

$$\nu(S) = \oplus_{i=1}^M \text{VAL}_u \left( \underset{\boldsymbol{\lambda} \in \boldsymbol{\Lambda}^S}{\arg\max} \, \text{VAL}_u(\boldsymbol{\lambda}, D_i), D_i \right) ,$$

*where $\oplus$ denotes an operator for aggregating the performances obtained for the individual datasets $D_i$, which for example can be instantiated by the mean, median, quantile, or interquartile mean. Hyperparameters $j \notin S$ that are not tuned as their domain is not contained in $\boldsymbol{\Lambda}^S$ are kept at their reference value $\lambda_j^0$. Consequently, the value of the empty coalition $S = \emptyset$ refers to the performance of the reference configuration, i.e., $\nu(\emptyset) = \oplus_{i=1}^M \text{VAL}_u(\boldsymbol{\lambda}^0, D_i)$.*

While this definition considers the problem of tunability across datasets, we can also question the usefulness of tuning the hyperparameters of a learner for a specific dataset at hand. This setting, which we dub Data-Specific Tunability, can be derived as a special instance of Definition 2:

**Definition 3** (HPI Game - Data-Specific Tunability). *Data-Specific Tunability is defined as a special case of Definition 4, considering a dataset collection of size $|\mathcal{D}| = 1$. Then, the value function can be simplified to*

$$\nu(S) = \text{VAL}_u \left( \underset{\boldsymbol{\lambda} \in \boldsymbol{\Lambda}^S}{\arg\max} \, \text{VAL}_u(\boldsymbol{\lambda}, D), D \right) .$$

We note that, in practice, for evaluating a single coalition in Definition 2, $M$ HPO runs are carried out to approximate the $\arg\max$. In the specific case of Definition 3, we still need to conduct one HPO run per coalition evaluation. While this can result in considerable costs, we argue that using surrogate models that are, e.g., obtained through HPO via Bayesian optimization, can be used to simulate HPO runs, rendering HYPERSHAP more tractable.

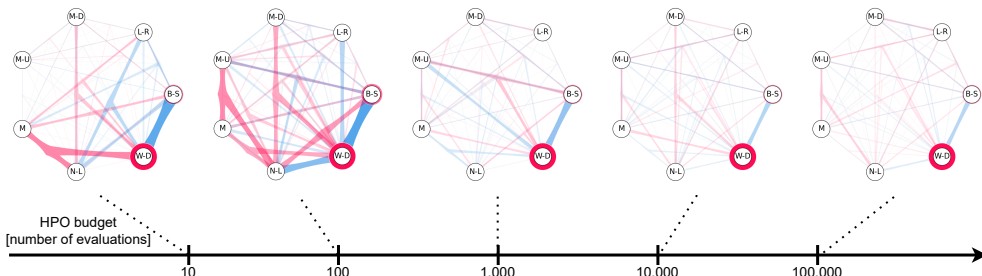

Figure 2: Hyperparameter importance with HyperSHAP, approximating the $\arg\max$ in Definition 3 of the value function via hyperparameter optimization with increasing budgets for dataset ID 7593 of `lcbench`. For tuning, we consider the following hyperparameters of `lcnet`: learning rate (L-R), batch size (B-S), weight decay (W-D), num layers (N-L), momentum (M), max units (M-U), and max dropout (M-D).

### 5.3 OPTIMIZER BIAS

In the previous section, we aimed to explain the importance of hyperparameters being tuned. However, depending on how much a hyperparameter may contribute to a performance gain can also be used to gain insights into the capabilities of a hyperparameter optimizer. More specifically, we would like to investigate whether a hyperparameter optimizer may fail to exploit certain hyperparameters. As shown in Figure 2, partial contributions of main effects and interactions to the overall performance vary depending on the approximation quality of the $\arg\max$. As a consequence, for high quality explanations, we need sufficiently accurate approximations. In turn, we may leverage this fact to detect deficiencies of optimizers. To this end, we define a hyperparameter optimizer to be a function $\mathcal{O}: \mathbb{D} \times 2^{\boldsymbol{\Lambda}} \to \boldsymbol{\Lambda}$, mapping from the space of datasets and a (sub)space of a hyperparameter configuration space to a configuration.

**Definition 4** (HPI Game - Optimizer Bias). *The Optimizer Bias HPI game is defined as a tuple*

$$G_O = (\mathcal{N}, \boldsymbol{\Lambda}, \boldsymbol{\lambda}^0, \mathcal{O}, \mathcal{D}, \nu),$$

*where $\mathcal{N}, \boldsymbol{\Lambda}, \boldsymbol{\lambda}^0, \mathcal{D}$, and the construction of $\boldsymbol{\Lambda}^S$ are as in Definition 2, $\mathcal{O}$ the hyperparameter optimizer of interest, and a value function $\nu$. Then, the value function is defined as*

$$\nu(S) = \oplus_{i=1}^{M} \left[ \text{VAL}_u \left( \mathcal{O}(D_i, \boldsymbol{\Lambda}^S), D_i \right) - \text{VAL}_u \left( \arg\max_{\boldsymbol{\lambda} \in \boldsymbol{\Lambda}^S} \text{VAL}_u(\boldsymbol{\lambda}, D_i), D_i \right) \right] .$$

Intuitively speaking, the value function measures any deviation from the performance of the actual best-performing hyperparameter configuration. In other words, with the help of Definition 4, we can identify deficiencies of the hyperparameter optimizer $\mathcal{O}$ over the actual best-performing solution and, thereby for example, identify whether an optimizer struggles to optimize certain (types of) hyperparameters.

However, to approximate the right-hand side of the difference, i.e., the $\arg\max$, in practice, we propose to employ a diverse ensemble of optimizers $\mathbb{O} := \{\mathcal{O}_i\}$. Furthermore, for some hyperparameter configuration space $\boldsymbol{\Lambda}^S$, we pick the best result obtained through any optimizer from $\mathbb{O}$. Thereby, we obtain a virtual best hyperparameter optimizer:

$$\mathcal{VBO}(D_i, \boldsymbol{\Lambda}^S) \mapsto \hat{\boldsymbol{\lambda}}, \hat{\boldsymbol{\lambda}} \in \arg\max_{\boldsymbol{\lambda}^i = \mathcal{O}_i(D_i, \boldsymbol{\Lambda}^S)} \text{VAL}_u(\boldsymbol{\lambda}^i, D_i) .$$

Analogue to Definition 3, we can define Data-Specific Optimizer Bias:

**Definition 5** (HPI Game - Data-Specific Optimizer Bias). *Data-Specific Optimizer Bias is defined as a special case of Definition 4, considering a dataset collection of size $|\mathcal{D}| = 1$.*

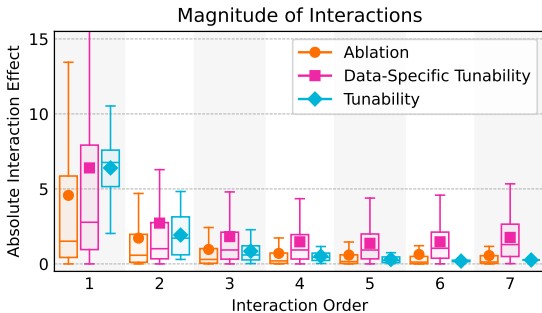 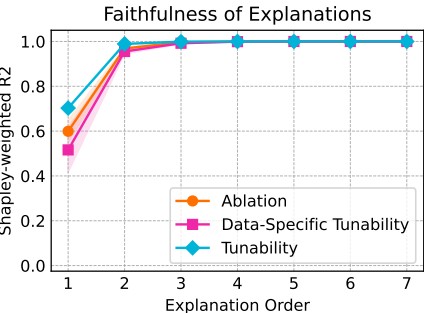

Figure 3: Interaction quantification for different HPI scenarios in terms of *magnitude* of interactions at each order (left) and *faithfulness* of lower order explanations (right, standard error shaded).

## 6 EXPERIMENTS

In this section, we showcase how the games proposed in Section 5 can be used to explain different aspects of the HPO pipeline with a particular focus on interactions between hyperparameters. To this end, we consider the benchmark suite `lcbench` (Zimmer et al., 2021) from the HPO benchmark package YAHPOGym (Pfisterer et al., 2022). Across three different levels, ranging from a specific configuration to the tunability of `lcnet` as the underlying network of `lcbench` to the analysis of optimizer biases, we demonstrate the usefulness of HYPERSHAP in Sections 6.1 and 6.2. Furthermore, in Section 6.3, we demonstrate how knowledge about hyperparameter importance can help to increase sample efficiency, but taking interactions into account is crucial. Details about the experiment setup, implementation, and additional results can be found in Appendix A and the technical supplement. For guidance on interpreting the SI visualizations, we kindly refer to Appendix B.

### 6.1 INTERACTIONS IN HYPERPARAMETER OPTIMIZATION

Using HYPERSHAP, we analyze the interactions at different levels of hyperparameter optimization, demonstrating that while higher-order interactions are present, they can be effectively captured with second-order explanations. The interaction quantification is summarized in Figure 3. Additional details are provided in Appendix C. For each HPI game — Ablation, Data-Specific Tunability, and Tunability — we determine the *magnitude* of interactions by measuring the MIs for each order (see Figure 3, left). The MIs show that all levels of hyperparameter optimization include both lower- and higher-order interactions (with box plots above zero), and that lower-order interactions are the most influential (showing the largest range for orders 1 and 2). Data-Specific Tunability exhibits the most higher-order interactions, indicating that there are dataset-specific nuances.

In addition, we examine how *faithfully* lower-order representations capture the higher-order effects and behavior of different games (see Figure 3, right). For each game setting, we compute the FSII $\Phi_k^{\text{FSII}}$ with explanation order $k$. We then approximate the game values for each hyperparameter subset $S \subseteq \mathcal{N}$ as $\hat{\nu}_k(S) := \sum_{L \subseteq S:|L| \leq k} \Phi_k^{\text{FSII}}(L)$. Finally, we compare the approximated game values with the ground truth as determined by considering the complete game and calculate an $R^2\left(\nu(S), \hat{\nu}_k(S)\right)$ loss weighted with the Shapley kernel (Lundberg & Lee, 2017; Tsai et al., 2023). This assesses how well the FSII describe the game behavior and cover higher-order interactions. An $R^2$ score of one indicates a perfect fit, meaning all interactions are covered with the given explanation. Figure 3, right, displays the $R^2$ scores all explanation orders $k$, and demonstrates that the behavior of hyperparameter optimization games cannot be explained without interactions. The SV ($k = 1$) yield a low $R^2$ score, whereas second-order interactions ($k = 2$) already achieve a near-perfect reconstruction error, outlining an interesting avenue for future research on exploiting this fact for efficient approximation.

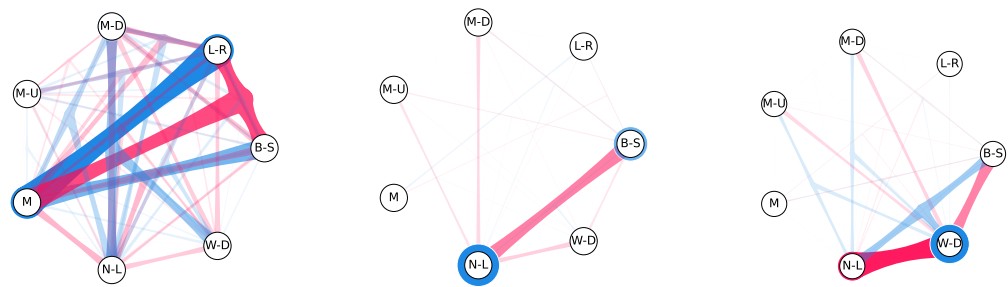

Figure 4: 3-SII plot for optimizing every hyperparameter individually (left) and for optimizing only a subset of hyperparameters consisting of learning rate (L-R), max dropout (M-D), and max units (M-U) in the center, and optimizing every hyperparameter but weight decay (W-D) on the right.

## 6.2 IDENTIFYING OPTIMIZER BIAS VIA HYPERPARAMETER IMPORTANCE

In this section, we consider Data-Specific Optimizer Bias as defined in Definition 5 to demonstrate how quantifying hyperparameter importance can be used to detect hyperparameter optimizer deficiencies. To this end, we implement two HPO strategies with built-in deficiencies. First, we consider a hyperparameter optimizer that optimizes every hyperparameter individually and composes the returned configuration from the individual best hyperparameter values, i.e., it ignores all potential interactions of hyperparameters. Second, we consider a hyperparameter optimizer that only optimizes a pre-defined subset of hyperparameters, resembling HPO methods that, e.g., can only tackle continuous hyperparameters and ignore categorical or integer hyperparameters. For the former, we expect the main effects of hyperparameters to be small but observe substantial negative interactions. The latter is expected to show small effects for the selected subset of hyperparameters and negative main effects for the rest.

As shown in Figure 4, we observe the effects expected for the two HPO strategies, validating our approach. While on the left, we analyze the hyperparameter optimizer tuning every hyperparameter independently, the main effects for the single hyperparameters are negligible (i.e., thin circles around the hyperparameters), demonstrating that the tuning of the individual hyperparameters works well. However, by design, the tuning behavior does not account for interactions between the hyperparameters, which results in large negative interactions, as indicated by the bold blue edges, red edges in turn indicate missing out on negative interactions. In the middle and on the right, we visualize Optimizer Bias for the HPO approach limiting to certain subsets of hyperparameters (see caption of the figure). We observe negligible main effects for the tuned hyperparameters, indicating close to optimal performance for tuning those. However, left-out hyperparameters are showing large main effects and also interactions with them are more pronounced, bringing the deficiencies of this HPO method to light. Momentum (M) and Weight Decay (W-D) show little change, reflecting their ineffectiveness for this dataset, and the subset tuner correctly avoids tuning them.

## 6.3 DOWNSTREAM TASK: IMPORTANCE-INFORMED HYPERPARAMETER OPTIMIZATION

In this section, we incorporate information obtained through HYPERSHAP in a subsequent HPO run. To this end, given a dataset $D$, we first determine the hyperparameter importance using HYPERSHAP. Based on this, we determine the top-2 hyperparameters with respect to their main effects, thus deliberately ignoring any kind of interaction these two hyperparameters may have with other hyperparameters (and themselves). We limit the hyperparameter configuration space to these two hyperparameters and conduct HPO runs (with information about hyperparameter importance) via random search, abbreviated as RS, (RS + HPI) (Bergstra & Bengio, 2012) and Bayesian optimization using SMAC (SMAC + HPI) (Hutter et al., 2011; Lindauer et al., 2022), a versatile and state-of-the-art package for HPO based on Bayesian optimization (Eggensperger et al., 2021). As baselines, we run both methods on the full hyperparameter configuration space, in the plots referred to as RS and SMAC, respectively, and compare their anytime performances for a budget of 100

**(a)** Dataset ID: 3945 (low interactivity)   **(b)** Dataset ID: 7593 (high interactivity)

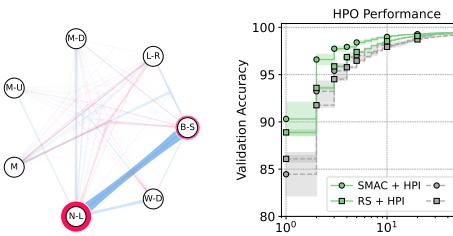
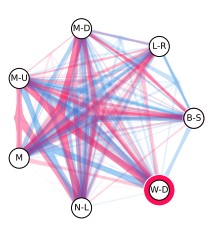
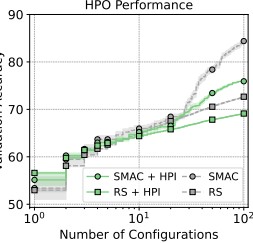

Figure 5: MI interaction graphs and anytime performances, showing mean accuracy $\pm$ standard error for dataset IDs 3945 (a) and 7593 (b). While in (a), the low level of interactions can be leveraged for shrinking the hyperparameter configuration space, in (b), a significant amount of interactions is present such that shrinking the hyperparameter configuration space deteriorates performance.

candidate evaluations. While we repeat SMAC runs 30 times with different random seeds, random search runs are repeated 1,000 times as random search is more noisy but cheaper to compute.

In Figure 5, we show interaction graphs for two datasets from lcbench (OpenML-IDs 3945, 7593) and anytime performance plots, plotting the number of evaluations on the $x$-axis versus accuracy on the $y$-axis. The graphs show the mean and the standard error is plotted in terms of error bands. We can observe that for Dataset 3945 where we have a relatively small level of interactions but strong main effects for the hyperparameters batch_size and num_layers, this information can be leveraged in both random search and SMAC. On the contrary, the presence of higher-order interactions, as visual for Dataset 7593, limiting to the top-2 hyperparameters yielding the highest main effects hurts anytime performance. As can be seen from the anytime performance plots, while having a slight advantage in the first few evaluations, considering the entire hyperparameter configuration space yields substantially better results within the total budget of 100 evaluations. From these observations, we conclude that taking higher-order interactions into account is crucial when being present, whereas if only a low level of interactions (and few major main effects) is detected, we can potentially leverage this information to speed up the HPO process. Recent results indicate that such scenarios with little interactions are more common in HPO than we might think (Pushak & Hoos, 2022) and thus can be quantified with the help of HYPERSHAP.

# 7   CONCLUSION & FUTURE WORK

In this paper, we proposed HYPERSHAP a comprehensive post-hoc explanation framework for quantifying hyperparameter importance via Shapley values and Shapley interactions on three levels: the values of hyperparameters, the tunability of hyperparameters, and the capabilities of hyperparameter optimizers. Compared to previous methods accounting for interactions between hyperparameters, HYPERSHAP attributes contributions to the performance rather than quantifying variance. While we can use HYPERSHAP to better understand the impact of hyperparameter values or tunability of hyperparameters, we also demonstrated that this knowledge can be immediately applied to downstream tasks. To this end, we showed how the anytime performance of a subsequent HPO run can benefit from focusing on hyperparameters identified as important.

In future work, we aim to extend the framework to the more general task of combined algorithm selection and hyperparameter optimization, and the design of entire ML pipelines (Olson & Moore, 2016; Wever et al., 2018; Heffetz et al., 2020; Feurer et al., 2022). In contrast to plain HPO, such more complex AutoML scenarios are less well studied, but with HYPERSHAP we now have a versatile and theory-grounded approach at hand that will allow a thorough study. Furthermore, we plan to develop methods for HPO that can leverage the information about the importance of hyperparameters. Learning hyperparameter importance across datasets could outline a promising direction to warmstart hyperparameter optimizers in an interpretable way and improve their efficiency based on past experience, leading to synergies with recent prior-guided HPO (Hvarfner et al., 2024) and human-centered AutoML (Lindauer et al., 2024).

## ETHICAL STATEMENT

In conducting this research on HyperSHAP, we have carefully considered the ethical implications of our work. This paper presents work with the goal to advance the field of machine learning (ML) and specifically the field of explainable artificial intelligence (XAI) and hyperparameter optimization (HPO). There are many potential societal consequences of our work. The aim of our study is to improve the transparency and interpretability of hyperparameter optimization, which is crucial for building trust and accountability into hyperparameter optimization methods. Thus, our research impacts a wide variety of ML application domains and therein can positively impact ML adoption and potentially reveal biases or unwanted behavior in HPO systems.

However, we recognize that the increased explainability provided by XAI also carries ethical risks. There is the potential for "explainability-based white-washing", where organizations, firms, or institutions might misuse XAI to justify questionable actions or outcomes. With responsible use, XAI can amplify the positive impacts of ML, ensuring its benefits are realized while minimizing harm.

## REPRODUCIBILITY STATEMENT

The code, datasets (in terms of pre-computed games), and hyperparameter configurations used in this work are stated in the experiment setup in Appendix A. Also the code provided as supplementary material will be made publicly available on GitHub upon acceptance. Clear instructions for reproducing our results are provided in the code supplement, including environment setup and dependency management. Detailed experimental results with random seeds are reported to ensure consistency in outcomes across different runs.

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

## ORGANISATION OF THE SUPPLEMENT MATERIAL

The technical supplement is organized as follows.

# A    EXPERIMENT SETUP

Our implementation builds upon the shapiq package (Muschalik et al., 2024a), which is publicly available on GitHub[1], for computing Shapley values and interactions. Furthermore, for the experiments, we use YAHPO-Gym (Pfisterer et al., 2022), a surrogate-based benchmark for multi-fidelity hyperparameter optimization. YAHPO-Gym provides several benchmark suites, i.e., `lcbench` (Zimmer et al., 2021), which we focused on in the main paper. However, in the subsequent sections, we also present results from the `rbv2_ranger` benchmark set, a random forest benchmark, from YAHPO-Gym demonstrating the more general applicability of HYPERSHAP. Furthermore, we run evaluations on the benchmark PD1 and JAHS-Bench-201 to showcase HYPERSHAP's wide applicability. In our repository, we provide pre-computed games to foster reproducibility of our results and allow for faster post-processing of the game values, e.g., for plotting different representations of the played games.

For better readability in terms of the font size, hyperparameter names are abbreviated in the interaction graphs.

**lcbench**  (Pfisterer et al., 2022; Zimmer et al., 2021) `lcbench` is a benchmark considering joint optimization of the neural architecture and hyperparameters that has been proposed by Zimmer et al. (2021) together with the automated deep learning system Auto-PyTorch. The benchmark consists of 35 datasets with 2000 configurations each for which the learning curves have been recorded, allowing for benchmarking multi-fidelity HPO. However, in YAHPO-Gym only 34 of the 35 original datasets are contained which is why our evaluation is also restricted to those 34 datasets.

| Hyperparameter Name | Abbreviation | Type |
|---|---|---|
| weight decay | W-D | float |
| learning rate | L-R | float |
| num layers | N-L | integer |
| momentum | M | float |
| max dropout | M-D | float |
| max units | M-U | integer |
| batch size | B-S | float |

**rbv2_ranger**  (Pfisterer et al., 2022) As already mentioned above, `rbv2_ranger` is a benchmark faced with tuning the hyperparameters of a random forest. We consider the hyperparameters of ranger as listed below:

| Hyperparameter Name | Abbreviation | Type |
|---|---|---|
| min node size | M-N | integer |
| mtry power | M-P | float |
| num impute selected cpo | N-I | categorical |
| num trees | N-T | integer |
| respect unordered factors | R-U | categorical |
| sample fraction | S-F | float |
| splitrule | S | categorical/Boolean |
| num random splits | N-R | integer |

**PD1**  (Wang et al., 2024) The PD1 benchmark is a testbed for evaluating hyperparameter optimization methods in the deep learning domain. It consists of tasks derived from realistic hyperparameter tuning problems, including transformer models and image classification networks. Across these different types of models, 4 hyperparameters are subject to tuning:

---

[1]https://github.com/mmschlk/shapiq

| Hyperparameter Name | Abbreviation | Type |
|---|---|---|
| lr_decay_factor | L-D | float |
| lr_initial | L-I | float |
| lr_power | L-P | float |
| opt_momentum | O-M | float |

**JAHS-Bench-201** (Bansal et al., 2022) To democratize research on neural architecture search, various table look-up and surrogate-based benchmarks have been proposed in the literature. Going even beyond plain neural architecture search, in JAHS-Bench-201, the combined task of searching for a suitable neural architecture and optimizing the hyperparameters of the learning algorithm is considered. We include it via the "'mf-prior-bench'" package that serves it with a surrogate model for predicting the validation error of a given architecture and hyperparameter configuration. The considered hyperparameters, including those for the neural architecture, are as follows:

| Hyperparameter Name | Abbreviation | Type |
|---|---|---|
| Activation | A | categorical |
| LearningRate | L | float |
| Op1 | Op1 | categorical |
| Op2 | Op2 | categorical |
| Op3 | Op3 | categorical |
| Op4 | Op4 | categorical |
| Op5 | Op5 | categorical |
| Op6 | Op6 | categorical |
| TrivialAugment | T | Boolean |
| WeightDecay | W | float |

## A.1 APPROXIMATION OF ARGMAX

As per Definitions 2 to 4, for every coalition $S$, we need to determine the $\arg\max$. However, the true $\arg\max$ is hard to compute, so we approximate it throughout our experiments. For the sake of implementation simplicity and unbiased sampling, to this end, we use random search with a large evaluation budget of 10,000 candidate evaluations. As the configurations are independently sampled, for evaluating a configuration, we simply blind an initially sampled batch of 10,000 hyperparameter configurations for the hyperparameters not contained in the coalition $S$ by setting their values to the default value. After blinding, the surrogate model provided by YAHPO-Gym is then queried for the set of hyperparameter configurations and the maximum observed performance is returned.

## A.2 COMPUTING OPTIMIZER BIAS

For the experiments considering the HPI game of Data-Specific Optimizer Bias, we designed three HPO methods that focus on different structural parts of the hyperparameter configuration space. For the hyperparameter optimization approach, tuning every hyperparameter individually, when considering a hyperparameter for tuning, we sampled 50 random values for every hyperparameter. For the hyperparameter optimizer focusing on a subset of hyperparameters, we allowed for 50,000 hyperparameter configurations. For the VBO, we employed the considered limited hyperparameter optimizer and a random search with a budget of 50,000 evaluations on the full hyperparameter configuration space. We chose larger HPO budgets for these experiments to immediately ensure the built-in deficiencies become apparent and reduce noise effects. Howevér, they might also already be visible with substantially smaller budgets.

## A.3 HARDWARE USAGE AND COMPUTE RESOURCES

Initial computations for lcbench and rbv2_ranger have been conducted on consumer hardware, i.e., Dell XPS 15 (Intel i7 13700H, 16GB RAM) and a MacBook Pro (M3 Max - 16C/40G, 128GB RAM). Overall computations took around 10 CPUd, highlighting HYPERSHAP being lightweight

when combined with surrogates. In the course of the reviewing process, we re-computed the games for Ablation and Data-Specific Tunability of lcbench and rbv2_ranger and added PD1 and JAHS-Bench-201. These computations have been conducted on a high-performance computer with nodes equipped with $2\times$ AMD Milan 7763 ($2 \times 64$ cores) and 256GiB RAM of which 1 core and 8GB RAM have been allocated to the computations for a single game. While the latter experiments amounted to 10.71 CPU days, in sum, the computations for this paper accumulate roughly 21 CPU days.

The average runtimes per benchmark and game are as follows (Table 2):

| Benchmark | $|\Lambda|$ | $|\mathcal{D}|$ | Runtime Ablation [s] | Runtime Data-Specific Tunability [s] | Runtime Tunability [s] |
|---|---|---|---|---|---|
| PD1 | 4 | 4 | 64.9±16.0 | 862.4±13.7 | - |
| JAHS | 10 | 3 | 123.7±4.4 | 30,406.7±4750.9 (8h26m) | - |
| LCBench | 7 | 34 | 4.8±0.4 | 357.3±3.1 | 10,713.4 (2h58m) |
| rbv2_ranger | 8 | 119 | 26.4±6.8 | 6,717±767.3 | - |

Table 2: Mean $\pm$ standard deviation of the runtimes on a single CPU per benchmark and game.

# B    GUIDANCE ON INTERPRETING INTERACTION VISUALIZATIONS

To visualize and interpret lower-, and higher-order interactions such as $k$-SII and FSII scores or MIs, we employ the SI graph visualization by Muschalik et al. (2024a;b). An exemplary 3-SII graph is represented in Figure 6. Visualizations as in Figure 6 can be interpreted as follows. Each individual player (e.g. hyperparameter) is represented as a node with connecting hyperedges representing the strength and direction of interactions. Akin to the well-established force plots (Lundberg & Lee, 2017), positive interactions are colored in red and negative interactions in blue, respectively. The strength of an interaction is represented by the size and opacity of the hyperedge. To reduce visual clutter, small interactions below a predefined absolute threshold may be omitted from the graph. Notably, first-order interactions (i.e. individual player contributions, or main effects) are represented by the size of the nodes.

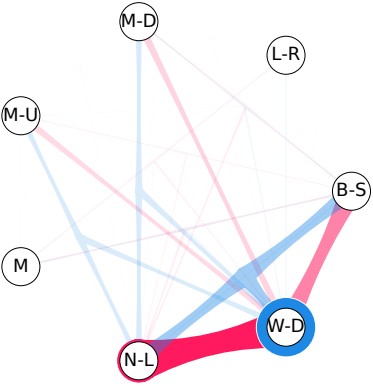

Figure 6: The 3-SII visualization replicated from Figure 4.

# C    ADDITIONAL
RESULTS FOR INTERACTION QUANTIFICATION

## C.1    MEASURING THE MAGNITUDE OF INTERACTIONS

In this section, we provide further details for measuring the presence of interactions discussed in Section 6.1. The MIs describe the pure additive effect of a coalition to the payout of the game. They thus serve as an important tool to analyze the interactions present in a game $\nu$. For instance, low-complexity games, where MIs are non-zero only up to coalitions of size $k$, are typically referred as $k$-additive games (Grabisch, 2016). In this case, SIs with explanation order $k$ perfectly recover all game values (Bordt & von Luxburg, 2023). In this case, the SIs correspond to the MIs. We thus analyze the absolute values of MIs for varying size of coalitions, i.e., displaying the strata $q(k) := \{|m(S)| : S \subseteq \mathcal{N}, |S| = k\}$ for varying interaction order $k = 1, \ldots, n$. Analyzing $q(k)$ indicates, if the game $\nu$ has lower- order higher-order interactions present by investigating the magnitudes and distributions in the strata $q(k)$.

## C.2    ANALYZING LOWER-ORDER REPRESENTATIONS OF GAMES

In this section, we provide additional details for the lower-order representations and $R^2$ scores discussed in Section 6.1. The SV that capture the fair contribution in a game $\nu$ of an individual to

the joint payout $\nu(\mathcal{N})$. However, as discussed in Section 4, the SV $\phi^{\text{SV}}(i)$ is also the solution to a constrained weighted least squares problem (Charnes et al., 1988; Fumagalli et al., 2024)

$$\phi^{\text{SV}} = \arg\min_{\phi} \sum_{T \subseteq \mathcal{N}} \frac{1}{\binom{n-2}{|T|-1}} \left( \nu(T) - \nu(\emptyset) - \sum_{i \in T} \phi(i) \right)^2 \text{ s.t. } \nu(\mathcal{N}) = \nu(\emptyset) + \sum_{i \in \mathcal{N}} \phi(i).$$

In other words, the SV is the best additive approximation of the game $\nu$ in terms of this weighted loss constrained on the efficiency axiom. Based on this result, the FSII (Tsai et al., 2023) was introduced as

$$\Phi_k^{\text{FSII}} := \arg\min_{\Phi_k} \sum_{T \subseteq N} \mu(|T|) \left( \nu(T) - \sum_{S \subseteq T, |S| \leq k} \Phi_k(S) \right)^2 \text{ with } \mu(t) := \begin{cases} \mu_\infty & \text{if } t \in \{0, n\} \\ \frac{1}{\binom{n-2}{t-1}} & \text{else} \end{cases},$$

where the infinite weights capture the constraints $\nu(\emptyset) = \Phi_k(\emptyset)$ and $\nu(\mathcal{N}) = \sum_{S \subseteq \mathcal{N}} \Phi_k(S)$. Note that Tsai et al. (2023) introduce FSII with a scaled variant of $\mu$ that does not affect the solution. The FSII can thus be viewed as the best possible approximation of the game $\nu$ using additive components up to order $k$ constrained on the efficiency axiom. It is therefore natural to introduce the *Shapley-weighted faithfulness* as

$$\mathcal{F}(\nu, \Phi_k) := \sum_{T \subseteq N} \mu(|T|) \left( \nu(T) - \sum_{S \subseteq T, |S| \leq k} \Phi_k(S) \right)^2.$$

Based on this faithfulness measure, the Shapley-weighted $R^2$ can be computed. More formally, we compute the weighted average and the total sum of squares as

$$\bar{y} := \frac{\sum_{T \subseteq \mathcal{N}} \mu(|T|) \nu(T)}{\sum_{T \subseteq \mathcal{N}} \mu(|T|)} \text{ and } \mathcal{F}_{\text{tot}} := \sum_{T \subseteq \mathcal{N}} \mu(|T|) \left( \nu(T) - \bar{y} \right)^2,$$

which yields the Shapley-weighted $R^2$ as

$$R^2(k) := R^2(\nu, \Phi_k) := 1 - \frac{\mathcal{F}(\nu, \Phi_k)}{\mathcal{F}_{\text{tot}}}.$$

In our experiments, we rely on FSII, since this interaction index optimizes the faithfulness measure $\mathcal{F}$ by definition. However, $k$-SII satisfies a similar faithfulness property (Fumagalli et al., 2024). Since the FSII is equal to the MIs for $k = n$, we have that $\mathcal{F}(\nu, \Phi_n) = 0$ due to the additive recovery property of the MIs. Hence, $R^2(n) = R^2(\nu, \Phi_n) = 0$ in this case. Clearly, the $R^2(k)$ scores are monotonic increasing in $k$ by definition of FSII. An $R^2(k) \approx 1$ indicates an almost perfect recovery of all game values. In our experiments, we have shown that higher-order interactions are present, but lower-order representations (low $k$) are mostly sufficient to achieve very high $R^2$ scores. This indicates that higher-order interactions are present but do not dominate the interaction landscape in our applications. For instance, a single isolated higher-order interaction would yield much lower $R^2$ scores (Muschalik et al., 2024a).

### C.3 INTERACTION QUANTIFICATION FOR HYPERPARAMETER OPTIMIZATION OF RANGER

This section contains additional results regarding the interaction quantification. In addition to the analysis of `lcbench` (see Section 6.1), we also assess the interactivity of `rbv2_ranger`. The results are summarized in Figure 7. Violin plots of the *magnitude* of interactions can be found in Figure 7. Surprisingly, the analysis of the three HPI games — Ablation, Data-Specific Tunability, and Tunability— of `rbv2_ranger` reveals a similar trend to `lcbench`. Both higher-, and lower-order interactions are present in all three HPI scenarios. While the largest range of interactions are of first order, interactions exist until the highest order. Akin to Section 6.1, these higher-order interactions can be *faithfully* represented with lower order explanations of order $k = 2$. Compared to the neural network based `lcbench` benchmark, first-order explanations recover a substantially higher $R^2$ score for the random forest based `rbv2_ranger` on both Tunability scenarios. This may indicate that independent tuning of parameters can be more effective with traditional machine learning models than for deep architectures.

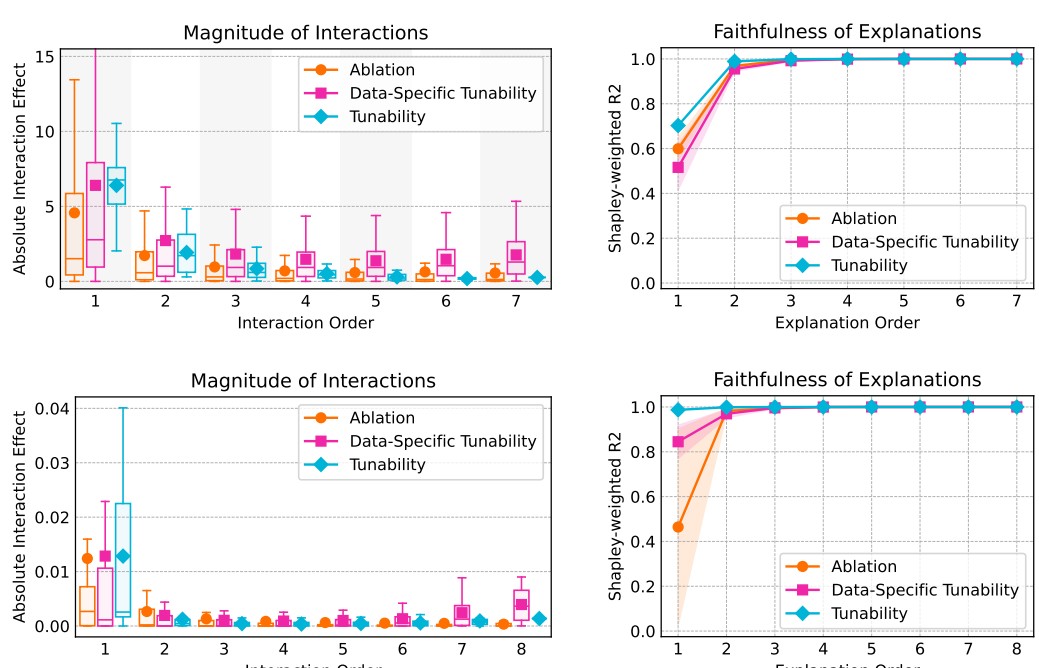

Figure 7: Interaction quantification for the three HPI scenarios, Ablation, Data-Specific Tunability, and Tunability in terms of *magnitude* of interactions at each order (left) and *faithfulness* of lower order explanations (right, standard error of the mean shaded). The first row shows the interaction quantification for `lcbench`, and the second row for `rbv2_ranger`.

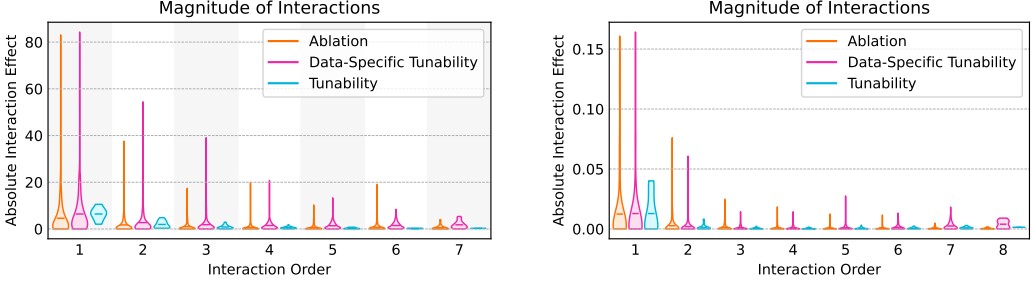

Figure 8: Violin plots of the *magnitude* for the three HPI scenarios, Ablation, Data-Specific Tunability, and Tunability for the `lcbench` (left) and `rbv2_ranger` (right).

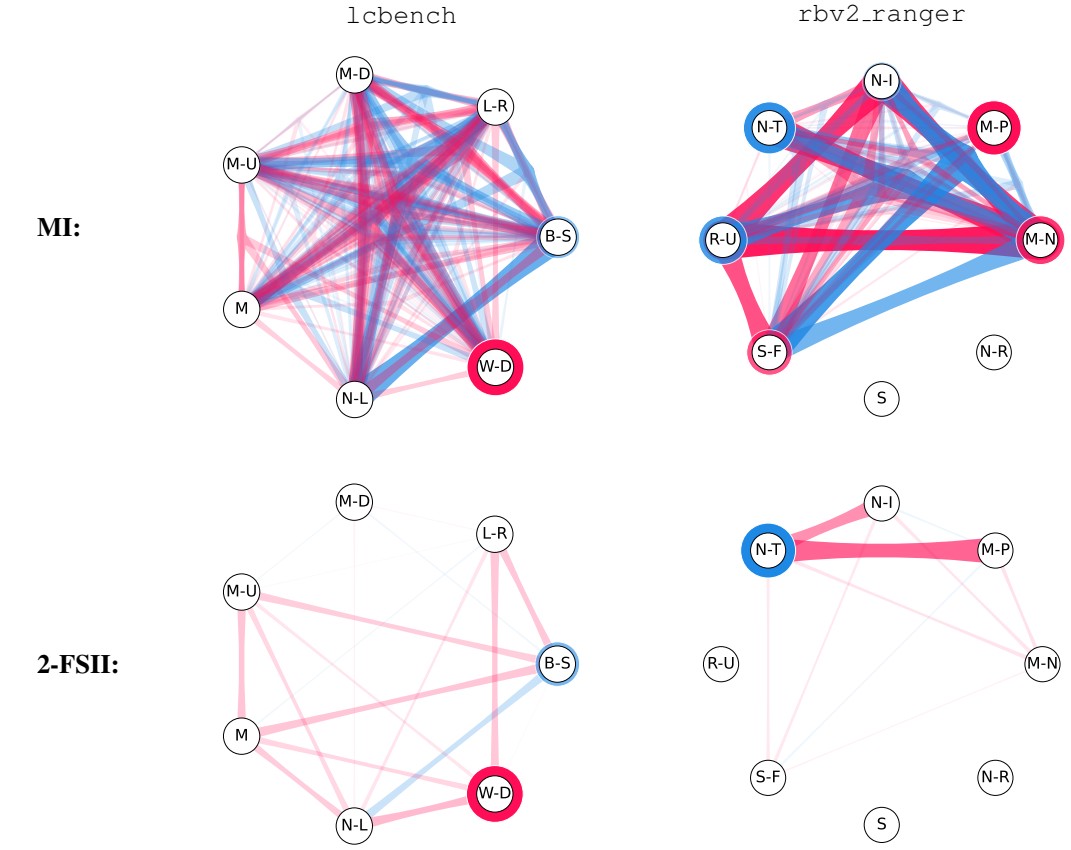

Figure 9: MIs and second order FSII interactions for the *Ablation* HPI Game for Dataset 7593 of `lcbench` (left) and Dataset 40981 of `rbv2_ranger` (right).

## C.4 ADDITIONAL INTERACTION VISUALIZATIONS

In this section, we present further interaction visualizations for both `lcbench` and `rbv2_ranger` comparing explanations obtained through MI and 2-FSII for the HPI games Ablation (see Figure 9), Data-Specific Tunability (see Figure 10), and Tunability (see Figure 11).

## C.5 RESULTS FOR JAHS-BENCH-201

In Figure 12, we present Moebius interactions between the hyperparameters of JAHS-Bench-201 for three different datasets: CIFAR10, FashionMNIST, and Colorectal Histology.

**Ablation:** At first sight, for CIFAR10 and Colorectal Histology, there are plenty of interactions, most of them between the different operations of the neural architecture. Hyperparameters for configuring the learning algorithm seem to have dense interactions with the operations chosen, showing the need to adjust the hyperparameters to the corresponding architecture. However, in the case of FashionMNIST, most of the performance gain seems to stem from configuring the learning rate together with the weight decay hyperparameter, indicating that the architecture we have optimized before running the Ablation HPI game does not yield too much of a performance improvement - furthermore, the learning rate seems to be changed for the worse in the considered HPC.

**Data-Specific Tunability:** The MIs for the Data-Specific Tunability show a significant amount of interactions between almost all hyperparameters and operations for the neural architecture. Across the three datasets, the operations typically show the largest main effects and also the strongest in-

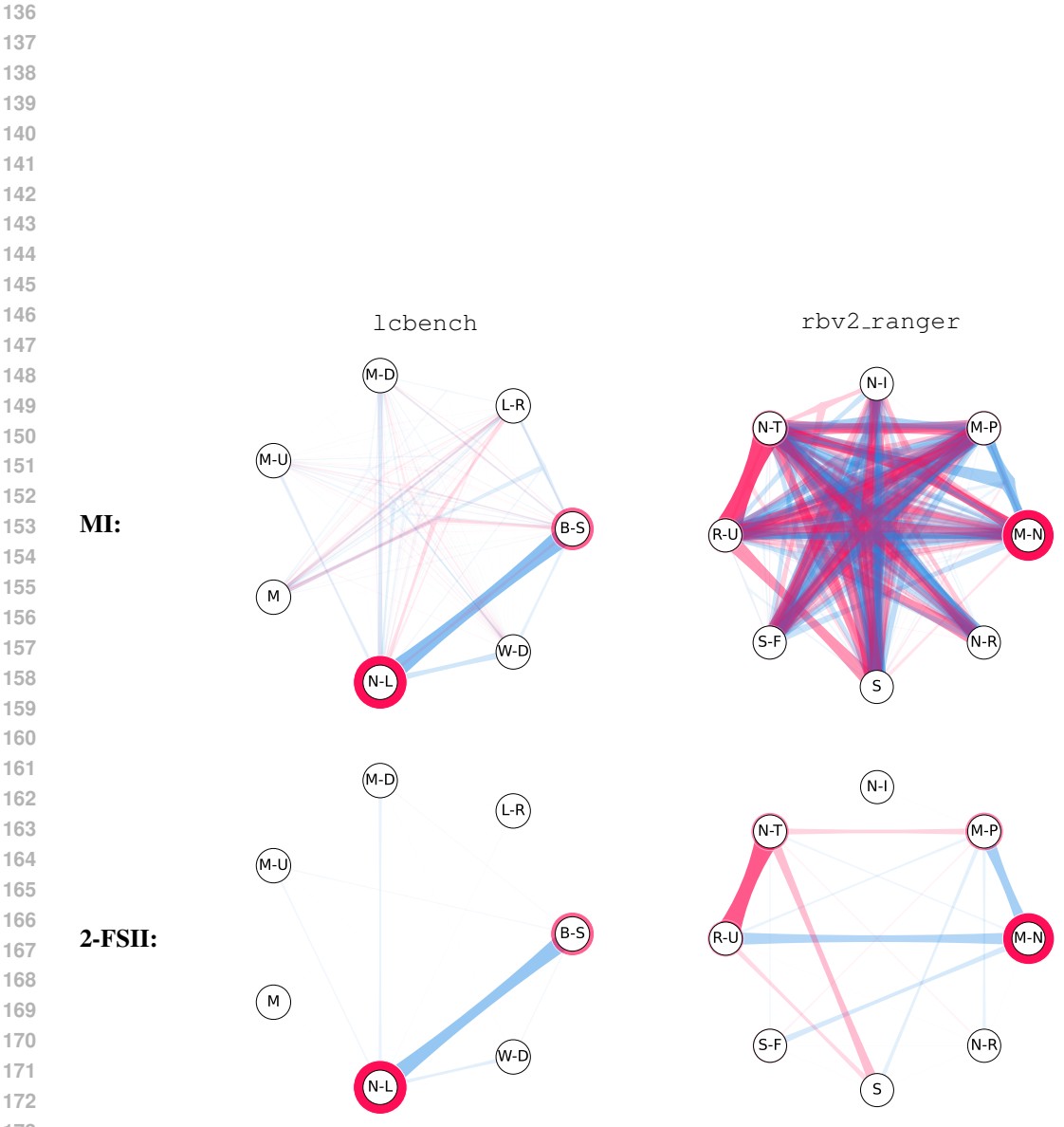

Figure 10: MIs and second order FSII interactions for the *Data-Specific Tunability* HPI Game for Dataset 3945 of `lcbench` (left) and Dataset 4135 of `rbv2_ranger` (right).

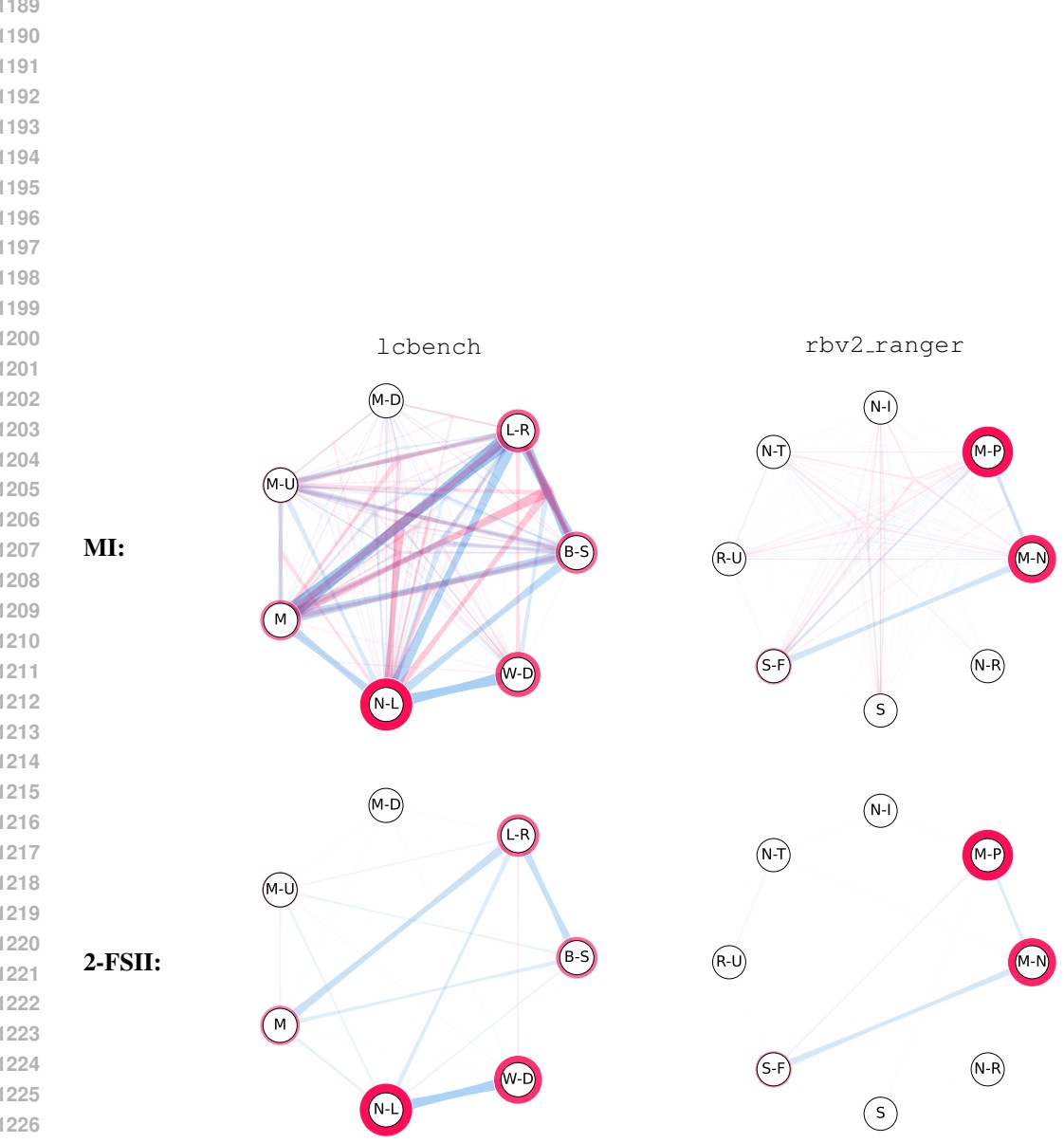

Figure 11: MIs and second order FSII interactions for the *Tunability* for lcbench (left) and rbv2_ranger (right).

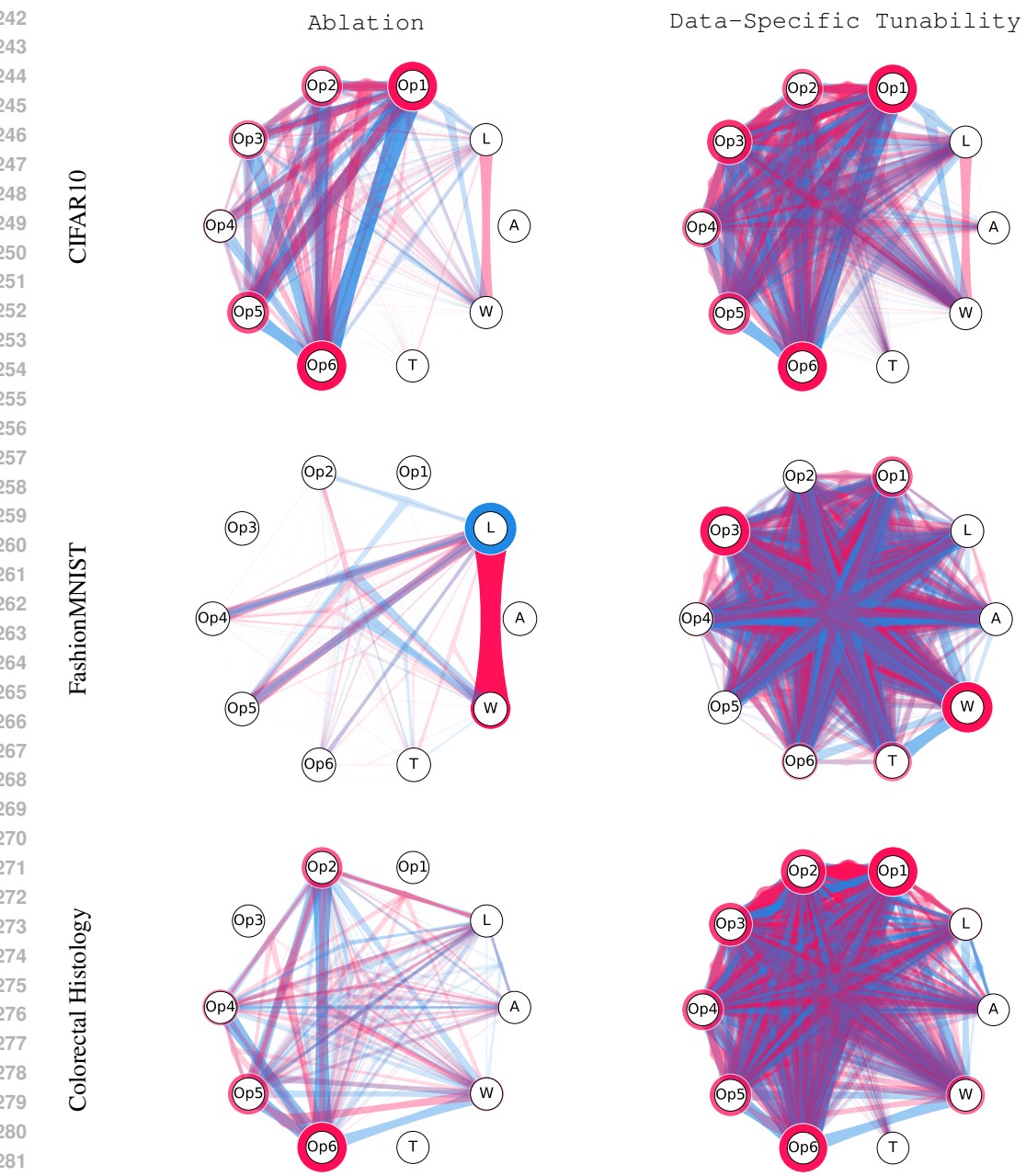

Figure 12: MIs for the two HPI games Ablation and Data-Specific Tunability on JAHS-Bench-201.

teractions among each other, which would also be expected, but also, the hyperparameters for the learning algorithm have a high degree of interactions, suggesting a need for joint optimization.

## C.6  RESULTS FOR PD1

In Figure 13, we present Moebius interactions between the hyperparameters of the PD1 benchmark with Ablation games on the right and Data-Specific Tunability on the right side of the figure. In the plots, we can observe that throughout the different scenarios and games, the initial learning rate has the largest impact on the overall performance of the learning procedure. In Data-Specific Tunability also obj_momentum appears as an important hyperparameter that has a strong negative interaction with the initial learning rate, indicating redundancy, i.e., optimizing one of the two hyperparameters

likely suffices to obtain the best possible performance. This is also reflected on the right-hand side of the figure, where we see that most of the performance improvement over the reference HPC stems from the initial learning rate. obj_momentum, in turn, is assigned almost no importance as the hyperparameter optimization process apparently focused on focusing on the initial learning rate.

# D  ADDITIONAL RESULTS FOR THE DOWNSTREAM TASK

In this section, we present the results of additional experiments using hyperparameter importance for informing downstream hyperparameter optimization tasks. To this end, we again select the top-2 hyperparameters according to main effects. In Figure 14 to 17 the results are visualized with the help of interaction graphs, violin plots showing Moebius coefficients for different orders of interaction, and the anytime performance of the hyperparameter optimizers for a budget of 100 candidate evaluations. Typically, the interaction graphs are normalized per dataset to account for effects on different scales. Here, in contrast, we normalize all interaction graphs jointly with the overall minimum and maximum to make the interaction graphs comparable to each other.

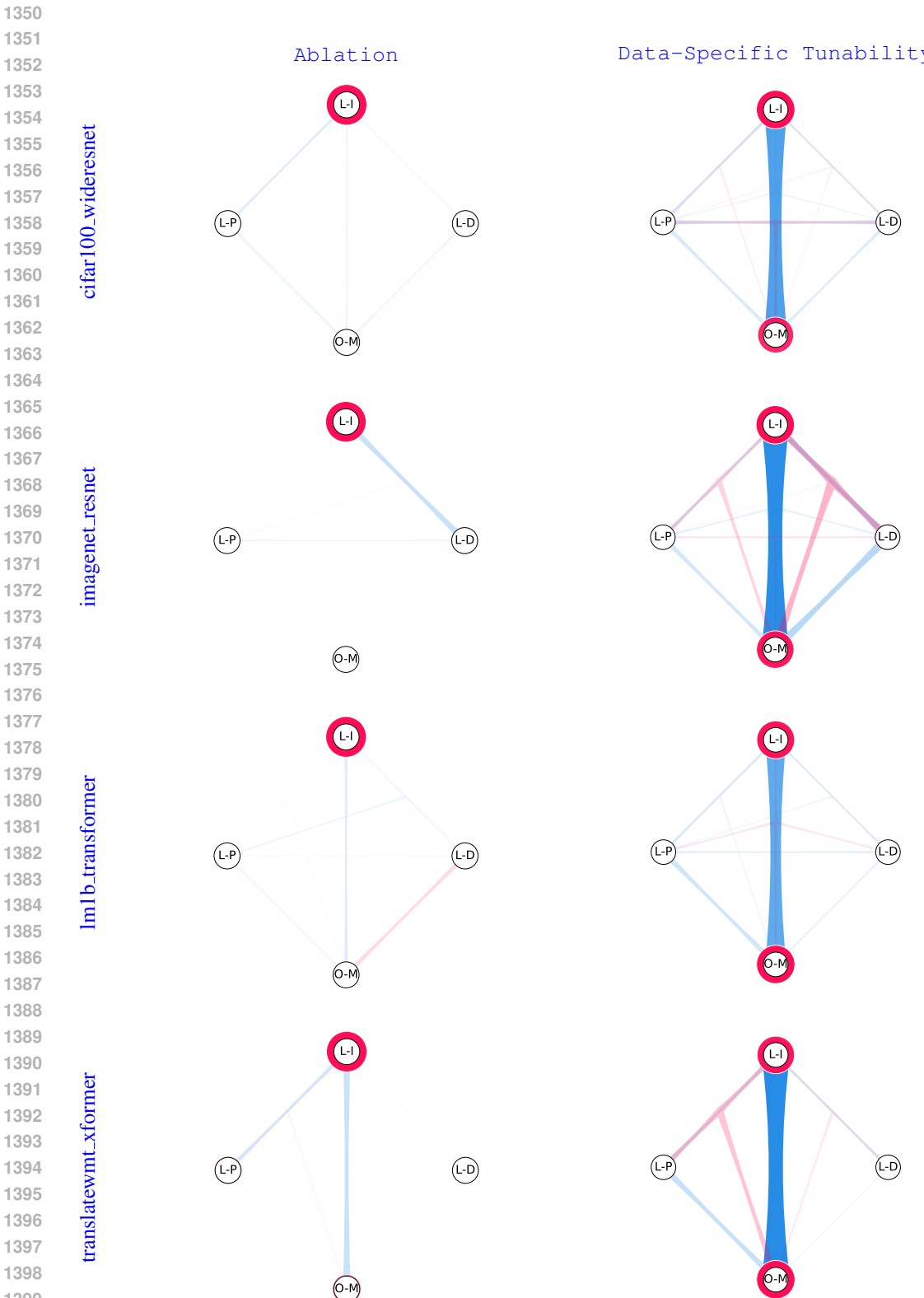

Figure 13: MIs for four different scenarios of the PD1 benchmark, considering hyperparameter optimization for Resnets and transformers.

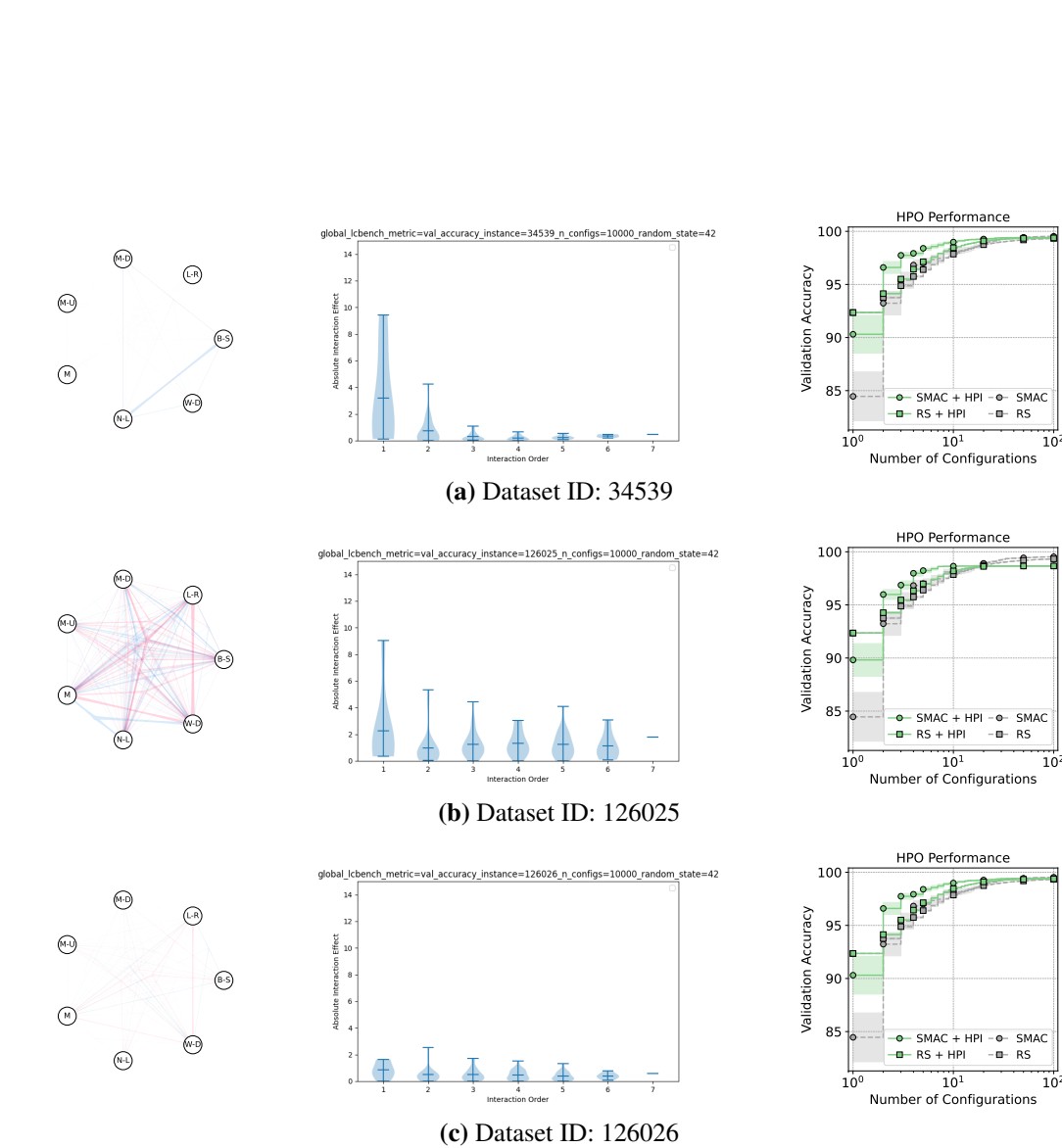

(a) Dataset ID: 34539

(b) Dataset ID: 126025

(c) Dataset ID: 126026

Figure 14: Downstream task HPO on `lcbench` benchmark for different datasets. On the left, interaction graphs are shown, visualizing the main effects of and interactions between hyperparameters. In the center column, the absolute amount of interactions is plotted for every order of interactions. On the right, anytime performance plots are shown for RS (+HPI), and SMAC (+HPI).

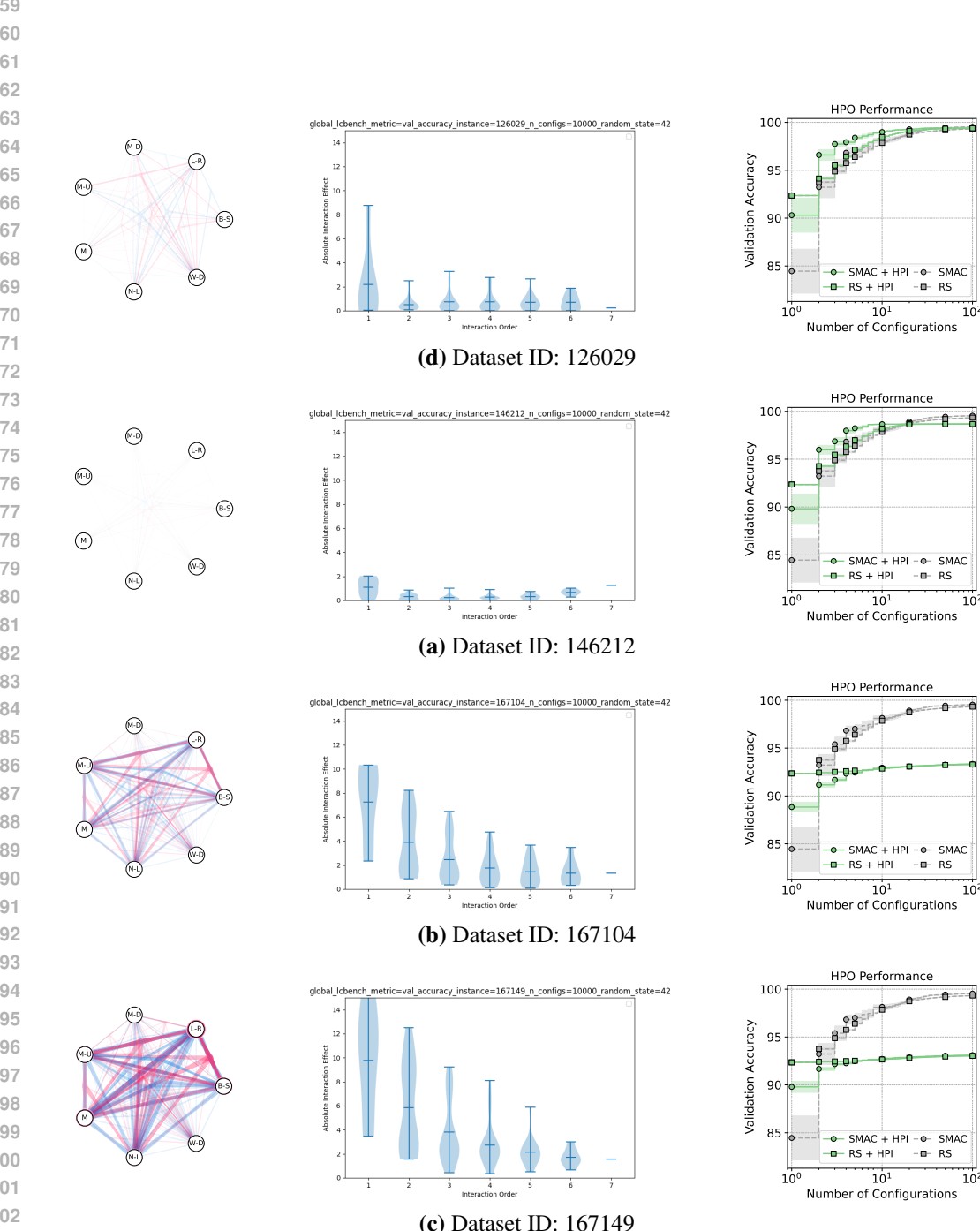

**(d)** Dataset ID: 126029

**(a)** Dataset ID: 146212

**(b)** Dataset ID: 167104

**(c)** Dataset ID: 167149

Figure 15: Downstream task HPO on `lcbench` benchmark for different datasets. On the left, interaction graphs are shown, visualizing the main effects of and interactions between hyperparameters. In the center column, the absolute amount of interactions is plotted for every order of interactions. On the right, anytime performance plots are shown for RS (+HPI), and SMAC (+HPI).

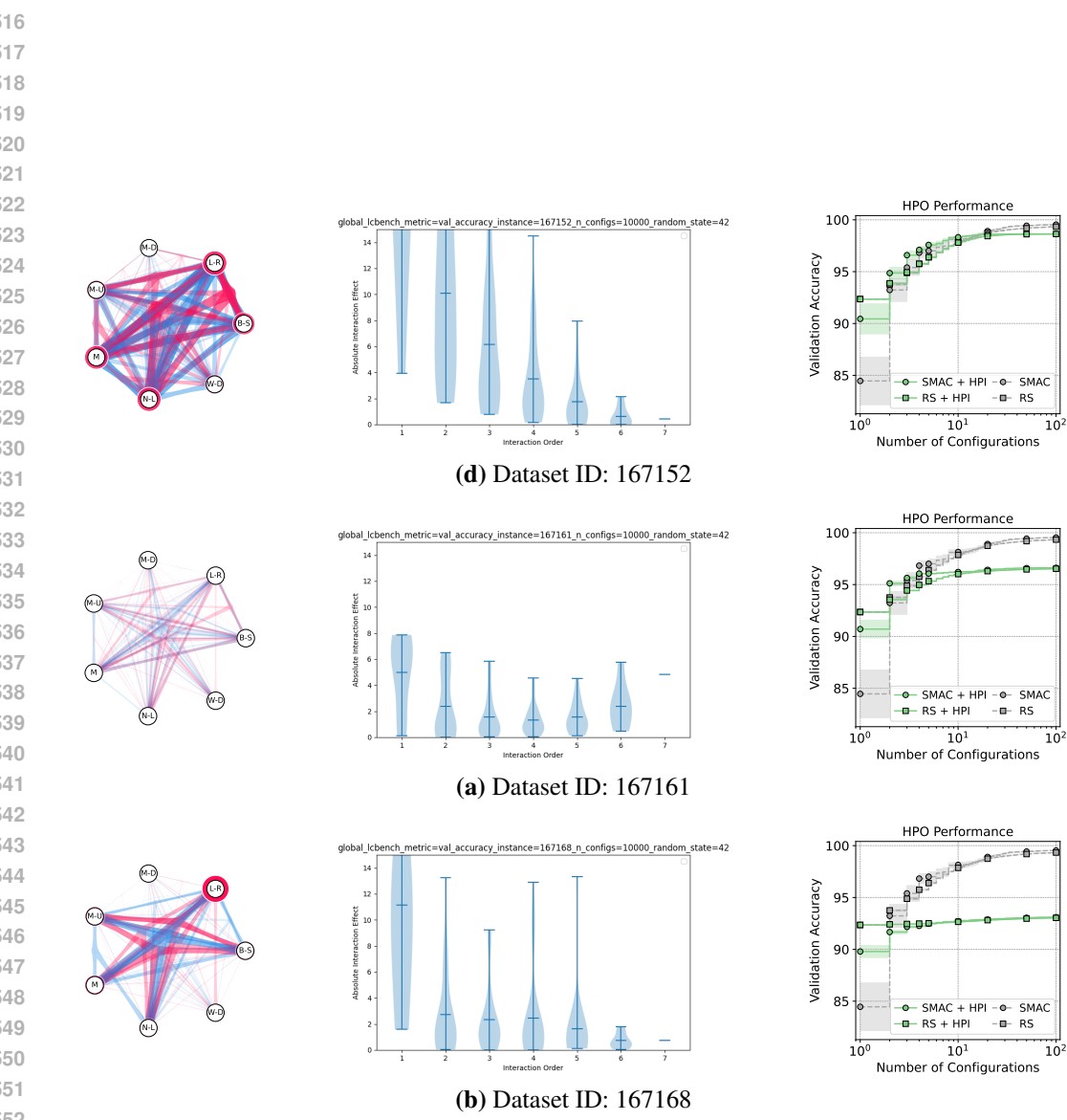

**(d)** Dataset ID: 167152

**(a)** Dataset ID: 167161

**(b)** Dataset ID: 167168

Figure 16: Downstream task HPO on `lcbench` benchmark for different datasets. On the left, interaction graphs are shown, visualizing the main effects of and interactions between hyperparameters. In the center column, the absolute amount of interactions is plotted for every order of interactions. On the right, anytime performance plots are shown for RS (+HPI), and SMAC (+HPI).

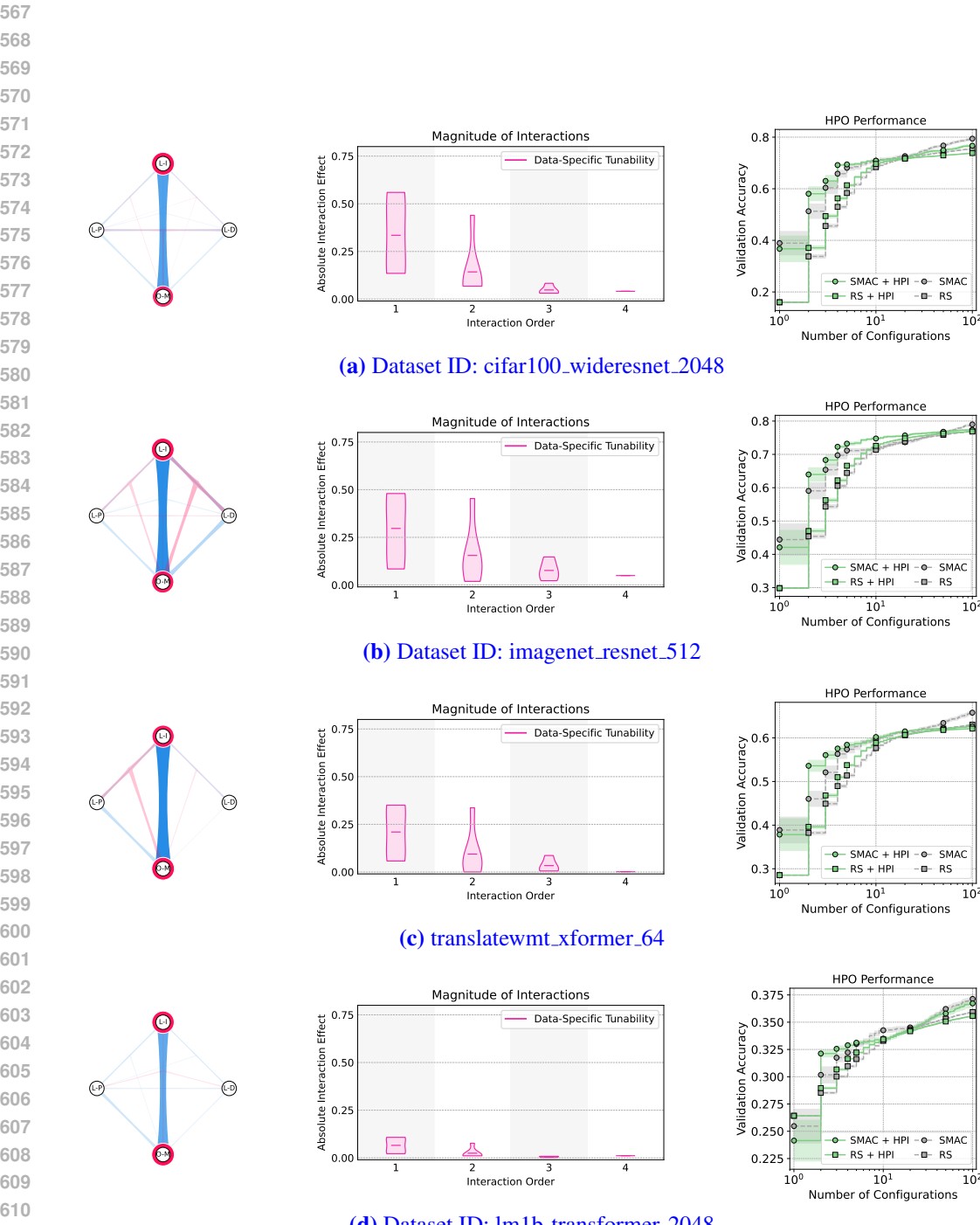

**(a)** Dataset ID: cifar100_wideresnet_2048

**(b)** Dataset ID: imagenet_resnet_512

**(c)** translatewmt_xformer_64

**(d)** Dataset ID: lm1b_transformer_2048

Figure 17: Downstream task HPO on `pd1` benchmark for the four different scenarios. On the left, interaction graphs are shown, visualizing the main effects of and interactions between hyperparameters. In the center column, the absolute amount of interactions is plotted for every order of interactions. On the right, anytime performance plots are shown for RS (+HPI), and SMAC (+HPI).

