# OpenReview forum: "HyperSHAP: Shapley Values and Interactions for Hyperparameter Importance"
_ICLR.cc/2025/Conference — Submitted to ICLR 2025_

### Official Review · Reviewer_6XF7 · 2024-10-22

**Soundness:** 2
**Presentation:** 2
**Contribution:** 1
**Rating:** 3
**Confidence:** 4

**Summary:**

This paper introduces HyperSHAP, a method for explaining hyperparameter importance and interactions using Shapley values and Shapley interaction indices from game theory. HyperSHAP analyzes hyperparameter importance at three levels: hyperparameter configuration, hyperparameter search space, and hyperparameter optimizer behavior. Through experiments on a downstream task, the paper claims the value of focusing on the important hyperparameters identified by this method.

**Strengths:**

- Explaining hyperparameter importance using Shapley values and Shapley interaction indices appears to be an unexplored approach.
- The paper provides sufficient background explanations for Shapley values and Shapley interaction indices.

**Weaknesses:**

- To demonstrate the practical usefulness of HyperSHAP, its time cost should be clearly presented and compared.
  - As stated in the paper, Definition 2-4 require multiple HPO runs for argmax operations. Even with the use of an approximation, the time burden of HyperSHAP can still be significant.
  - The time burden of HyperSHAP raises concern about whether applying HyperSHAP before other HPO algorithms provides any real benefit.
  - More extensive and fair comparisons of HPO algorithms, with and without HyperSHAP, are necessary. Additionally, the time cost of HyperSHAP should be clearly reported.

- The paper does not include any comparisons with other methods specifically designed to measure hyperparameter importance or interactions.

- For HyperSHAP to be a reliable interpretability technique, its results on hyperparameter importance and interactions should match real benchmarks.
  - The experiments in Section 6.2 are quite limited in scope, analyzing only two types of toy optimizers on a single dataset.

**Questions:**

Please refer to the Weaknesses section.

---

### Official Review · Reviewer_GqNM · 2024-10-29

**Soundness:** 3
**Presentation:** 2
**Contribution:** 3
**Rating:** 6
**Confidence:** 3

**Summary:**

This paper introduces HyperSHAP, a novel framework that utilizes Shapley values from cooperative game theory to assess and interpret hyperparameter importance and interactions in machine learning models. HyperSHAP provides a structured approach to hyperparameter optimization (HPO) by defining three "hyperparameter importance (HPI) games" to evaluate the impact of individual hyperparameters and their interactions. The framework supports three levels of analysis: (1) Ablation, which compares performance changes by switching individual hyperparameters; (2) Tunability, assessing the potential for performance improvement via tuning; and (3) Optimizer Bias, examining any systematic biases in hyperparameter optimization. HyperSHAP's efficacy is demonstrated through experiments that highlight its potential to improve HPO efficiency and enhance model performance by focusing on influential hyperparameters.

**Strengths:**

* The framework targets explainable hyperparameter optimization, a critical aspect of machine learning that directly impacts model performance and efficiency, and that is usually important for practitioners.
* HyperSHAP introduces Shapley values in a unique way to evaluate hyperparameters, capturing both individual and interaction effects, thus contributing to a more comprehensive understanding of hyperparameter importance.
* The results validate HyperSHAP’s utility, showing that focusing on high-impact hyperparameters the framework identifies can lead to better model performance and optimization efficiency.

**Weaknesses:**

* The experiments were limited in terms of search spaces and data modalities. Testing HyperSHAP on diverse configurations would improve its generalizability and robustness.
* The study would benefit from a comparison with other baseline explainable HPO techniques to assess HyperSHAP’s relative performance and computational efficiency.
* Calculating Shapley values and interaction indices can be computationally intensive. Including performance and scalability insights would be helpful, especially for real-world applications where computational cost is a concern.
* The paper is difficult to understand in some parts. For example, the Figure 2 is difficult to follow.

**Questions:**

* How expensive in time and resources is to compute the shape values?

---

### Official Review · Reviewer_hK6m · 2024-10-31

**Soundness:** 2
**Presentation:** 2
**Contribution:** 2
**Rating:** 5
**Confidence:** 4

**Summary:**

The paper proposes HyperSHAP, a game-theoretic framework for explaining hyperparameter importance and interactions during Hyperparameter Optimization (HPO) of ML models. The framework considers three instances where hyperparameter importance deserves to be assessed:
- Ablation: Importance of hyperparameters in ablation studies.
- Tunability: Importance of hyperparameters for a given optimizer. This can be used with several datasets to test the importance across different datasets. It corresponds to the classical HPO setting.
- Optimizer bias: Importance of hyperparameters when considering several optimizers.  Can assess weaknesses of a given optimizer.
Experiments show the benefits of HyperSHAP in an HPO setting by identifying the importance of individual hyperparameters and their higher-order interactions; and emphasizing optimizers' flaws.

**Strengths:**

- The authors clearly define three instances of application for HyperSHAP that corresponds to concrete situations that ML practitioners regularly encounter.
- The experiment effectively illustrates the claim that hyperparameters' importance can be different and that their interactions may impact the final validation error under different scenarios.
- The paper is well structured.

**Weaknesses:**

Even if the paper is well structured, it is difficult to follow :
- Much information that is required to understand what is going on is deferred to the appendix (e.g., how to read the plots, how the argmax are computed - there is not even a link to the corresponding appendix - for definitions 2 and 4). We should not have to resort to the Appendix to understand the results presented.
- l.421. All of a sudden, the authors use FSII, whereas it was barely mentioned earlier, and we do not know what it is.
- I find that the take-aways of the experiments are not clearly stated in Section 6
- Sometimes the link between the conclusions and the experiment's results is not clear. For instance
	- l.348 "For lower budgets, weight decay is not considered important." really? It seems equally important regardless of the number of evaluations in Figure 2.
	- l.350 "the optimizer leverages the strong positive interaction between weight decay and learning rate" There never seem to be any interaction between those two in Figure 2.

Although the framework identifies that hyperparameters can be of different importance, as well as their interaction, I am not convinced of its practical utility.
- The computational budget of this study is a serious concern for me. Indeed, evaluating SI involves a combinatorial number of model evaluations. The authors specify (in the appendix - it should be in the manuscript in my opinion) that they are evaluated with thousands of random evaluations, but it has a cost that is not taken into account e.g. in Section 6.3 in the x-axis reporting the number of evaluations. As a result, it is not sure that it effectively enhances the overall efficiency of HPO.
- What is the point of evaluating the weaknesses of a given optimizer if it implies to evaluate $\underset{\lambda}{\operatorname{argmax}} VAL_u(\lambda, D_i)$ l.360, i.e. basically solving the problem of HPO?

**Questions:**

Can you provide an updated comparison of RS and SMAC vs RS + SI and SMAC + SI with the same computational budget?

---

### Official Review · Reviewer_1AL6 · 2024-11-03

**Soundness:** 3
**Presentation:** 3
**Contribution:** 3
**Rating:** 6
**Confidence:** 3

**Summary:**

The paper proposes a framework that uses Shapley values and interactions to quantify the importance of hyperparameters. It starts with a detailed description of what the Shapley values and interactions are, and then uses these in the context of hyperparameter optimization to define the HyperSHAP framework. It includes an experimental evaluation on LCBench to show how it can be used and how it can lead to obtaining stronger performance by suitably reducing the search space based on the identified hyperparameter importances.

**Strengths:**

* The paper introduces a new well-motivated strategy to quantify the importance of hyperparameters, including their interactions. The strategy uses Shapley values and interactions for this goal and there is relatively extensive contribution in terms of framing these in a way suitable for the problem of hyperparameter optimization.
* The paper is well-written with precise explanations and well-made figures to explain the findings.
* The paper shows how the framework gives support to finding that only lower-order interaction is
typically sufficient to explain most of the performance improvements when conducting hyperparameter optimization. This is in line with the mentioned literature and is good to confirm via HyperSHAP.
* The paper shows the framework can be utilized in practice to focus on important hyperparameters and reduce the search space, improving anytime performance in constrained budget settings.

**Weaknesses:**

To me the main weakness is the limited experimental evaluation of the framework. It is evaluated on selected datasets within LCBench for neural networks (with rbv2 ranger benchmark in the appendix for random forests). The experimental evaluation works mainly as a way to showcase how the framework can be used and that it can be successfully used in at least some cases.

It would be quite valuable to see a more extensive evaluation, via a larger number of benchmarks with diverse sets of hyperparameters used - with some summary findings reported. Some of these could focus on fine-tuning settings (e.g. of large language models) as these are often of interest. Various pre-computed benchmarks exist so such significantly more rigorous evaluation may be manageable to conduct. For example, the PD1 benchmark could be a suitable example. Such more rigorous evaluation would help better establish the usefulness of the proposed methodology.

PD1 benchmark: Pre-trained Gaussian Processes for Bayesian Optimization. Wang et al., JMLR’24
https://arxiv.org/pdf/2109.08215

**Questions:**

* Am I correct in understanding that the paper uses pre-computed benchmarks for evaluating the performance of sampled hyperparameters? I.e. it does not need to run long experiments with the sampled configurations.
* HPO is often studied also in the form of neural architecture search. Does the framework also apply to this setting and is interesting to use there? For example to find good architectures in benchmarks such as NAS-Bench-201.

NAS-Bench-201: Extending the Scope of Reproducible Neural Architecture Search. Dong et al., ICLR’20
https://openreview.net/forum?id=HJxyZkBKDr

---

### Meta-Review · Area_Chair_5yFe · 2024-12-24

**Metareview:**

This paper presents HyperSHAP, a method for explaining hyperparameter importance and interactions using Shapley values and Shapley interaction indices from game theory. The framework identifies three key scenarios for assessing hyperparameter importance: (1) Ablation, which evaluates the impact of hyperparameters in ablation studies; (2) Tunability, which measures the importance of hyperparameters for a given optimizer across multiple datasets, aligning with the classical HPO setting; and (3) Optimizer Bias, which examines hyperparameter importance across different optimizers to identify potential weaknesses. Experiments demonstrate the utility of HyperSHAP in HPO settings by identifying individual hyperparameter importance, higher-order interactions, and optimizer flaws.

The strengths of the paper include its novel application of Shapley values and interaction indices for explaining hyperparameter importance, as well as its clear and accessible writing. However, the work has notable weaknesses, including the lack of a fair comparison between HPI + HPO and HPO under the same computational budget (e.g., wall-clock time) and insufficient comparisons with other established HPI methods to validate its effectiveness and necessity.

Given these limitations, I recommend rejecting this submission.

**Additional Comments On Reviewer Discussion:**

During the discussion period, all reviewers actively engaged with the authors.

After carefully reviewing the concerns raised by the reviewers, I found that the authors failed to adequately address two critical issues highlighted by Reviewer hK6m and Reviewer 6XF7.

Specifically, Reviewer hK6m raised concerns about the fairness of the comparison between HPI + HPO and HPO, emphasizing that such comparisons should be made under the same computational budget. Unfortunately, the authors’ responses were not satisfactory, as the experiments in Section 6.3, Figure 5, were conducted under the same number of configurations rather than the same computational budget. The x-axis should represent wall-clock time, and considering the significant computational cost of HPI, it is likely that HPO would still outperform HPI + HPO under a fair comparison.

Reviewer 6XF7 also highlighted the lack of comparisons with existing HPI methods, which are crucial to validate the effectiveness and necessity of introducing another HPI method, such as HyperSHAP. Unfortunately, the authors chose to disregard this essential request.

Additionally, Reviewer 1AL6 did not advocate for the acceptance of this work.

These unresolved issues severely undermine the validity and potential impact of the paper, limiting its relevance to the broader research community. Consequently, I believe the submission does not meet the high standards expected for acceptance at the prestigious ICLR conference.

---

### Decision · Program_Chairs · 2025-01-22

Reject